# Screen time and early adolescent mental health, academic, and social outcomes in 9- and 10- year old children: Utilizing the Adolescent Brain Cognitive Development SM (ABCD) Study

Katie N. Paulich[1,2]*, J. Megan Ross[1◉¤], Jeffrey M. Lessem[1◉], John K. Hewitt[1,2]

1 Institute for Behavioral Genetics, University of Colorado Boulder, Boulder, CO, United States of America,
2 Department of Psychology and Neuroscience, University of Colorado Boulder, Boulder, CO, United States of America

◉ These authors contributed equally to this work.
¤ Current address: University of Colorado Denver—Anschutz Medical Campus, Denver, CO, United States of America
* katie.paulich@colorado.edu

**Data Availability Statement:** Legal restrictions on sharing the utilized de-identified data set are in place; the data are owned by a third-party

## Abstract

In a technology-driven society, screens are being used more than ever. The high rate of electronic media use among children and adolescents begs the question: is screen time harming our youth? The current study draws from a nationwide sample of 11,875 participants in the United States, aged 9 to 10 years, from the Adolescent Brain Cognitive Development Study (ABCD Study®). We investigate relationships between screen time and mental health, behavioral problems, academic performance, sleep habits, and peer relationships by conducting a series of correlation and regression analyses, controlling for SES and race/ethnicity. We find that more screen time is moderately associated with worse mental health, increased behavioral problems, decreased academic performance, and poorer sleep, but heightened quality of peer relationships. However, effect sizes associated with screen time and the various outcomes were modest; SES was more strongly associated with each outcome measure. Our analyses do not establish causality and the small effect sizes observed suggest that increased screen time is unlikely to be directly harmful to 9-and-10-year-old children.

## Introduction

Children and adolescents are spending more time on screens and electronic media than ever before, with 95% of teens in the United States having access to a smartphone [1]. While global inequalities in technology use certainly exist—in 71 out of 195 countries globally, less than half the population has access to the internet—it is undeniable that average global technology use is on the rise, especially among youth [2]. With the rise in media use, one might ask whether

**Funding:** Data used in the preparation of this paper were obtained from the Adolescent Brain Cognitive Development (ABCD) Study (https://abcdstudy.org), held in the National institute of Mental Health Data Archive (NDA). This is a multisite, longitudinal study designed to recruit more than 10,000 children age 9-10 and follow them over 10 years into early adulthood. The ABCD Study is supported by the National Institutes of Health (NIH) and additional federal partners under award numbers U01DA041022, U01DA041028, U01DA041048, U01DA041089, U01DA041106, U01DA041117, U01DA041120, U01DA041134, U01DA041148, U01DA041156, U01DA041174, U24DA041123, and U24DA041147. A full list of supporters is available at https://abcdstudy.org/nih-collaborators. A listing of participating sites and a complete listing of the study investigators can be found at https://abcdstudy.org/principal-investigators.html. ABCD consortium investigators designed and implemented the study and/or provided data but did not necessarily participate in analysis or writing of this report. This manuscript reflects the views of the authors and may not reflect the opinions or views of the NIH or ABCD consortium investigators. The ABCD data repository grows and changes over time. The ABCD data used in this paper came from [NIMH Data Archive Digital Object Identifier (10.15154/1519271)].

**Competing interests:** The authors have declared that no competing interests exist.

screen time—the amount of time one spends on electronic media, usually viewing electronic screens—is harming youth. Adolescence is a critical developmental period [3] during which important aspects of health and well-being are easily influenced. As electronic media use among adolescents climbs, screens are increasingly incorporated into adolescents' development [4] and, therefore, potential relationships between screen time and adolescent well-being are of interest. Among the most important markers of adolescent well-being are internalizing and externalizing disorders [5], academic performance [6], sleep [7], and peer relationships [8].

Previous literature links increased screen time to a number of negative outcomes, including poor mental health and worse behavioral problems [9]; internalizing problems during adolescence have been linked with impaired development of autonomy, identity, morality, and social responsibility [10]. One study thus far has examined the relationship between screen time and mental health in the context of anxiety and depression in 4528 participants from an early data release of the Adolescent Brain Cognitive Development Study (ABCD Study), reporting that, when controlling for participant age, sex, BMI, family income, race, and physical activity, screen time was positively associated with anxiety and depression [11]. The authors found that child report of weekday electronic media use and both child and parent report of weekend electronic media use were significantly associated with anxiety, and both child and parent report of both weekend and weekday electronic media use were significantly associated with depression. The same study also found that different types of screen time (e.g., television, texting) showed differential relationships to anxiety and depression. Another study has linked screen time to increased adolescent depression and anxiety diagnoses, as well as to prevalence of treatment by a mental health professional and subsequent use of medication for psychological or behavioral health concerns [12].

In addition to internalizing problems, screen time has been linked to externalizing behavioral concerns (e.g., aggression). One publication examined technology-related parenting strategies, reporting that strategies with increased child screen time were associated with more externalizing behavioral problems [13]. Nevertheless, externalizing problems are the most common and persistent problem behaviors seen in childhood and adolescence [14], and therefore, should be examined in studies focused on adolescent development. Screen time has also been associated with attention problems; one study utilized a sample of French students to examine the relationship between self-report screen time and self-perceived attention problems, including attention-deficit hyperactivity disorder (ADHD). Researchers found a significant association between screen time and total score of self-perceived attention problems and hyperactivity levels [15].

There has also been a documented decline in academic performance with increased screen time [16], which may have implications for overall grade point average and potential college admission. When clustering screen time and sleep time, researchers saw that participants with higher academic achievement scores tended to spend less time on screens and sleep more than their peers with lower academic achievement scores; previous studies have also demonstrated negative associations between screen time and sleep quantity and quality [17], such that those who spend more time on screens get less sleep overall and more interrupted sleep. Because of the relationship between sleep deficits and mood and cognitive problems [18], sleep has a direct impact on adolescent well-being. Another study used the ABCD Study sample in examining the mediating role of sleep in screen time and problem behaviors in children, and found that sleep duration mediates the association between screen time and problem behaviors [19].

Formation of peer relationships is one of the most important and influential aspects of adolescence [20]. It has been shown that spending more time on screens is positively associated with the quality of peer relationships in school-age children [21]. Researchers reported that TV

and computer use—including social media and messaging—were related to more positive peer relationships, with the suggestion that those forms of media use might be culturally linked to socialization.

However, a recent annual research review reported that previous literature examining screen time has produced mixed results, and that screen time itself may not be cause for concern; rather, *how* electronic media is being consumed by adolescents is the more important consideration [22]. An Australian study also suggests that the type of screen time matters [23], and it has been posited that youth from low SES backgrounds may disproportionally impacted by psychological problems linked to high screen time [24].

Given the mixed results of previous research, and the importance of adolescence as a developmental period, investigation of relationships between screen time and those important aspects of adolescent development could shed new light on cornerstones of adolescent well-being. Additionally, if we were to identify that screen time may be problematic in areas of adolescent development, there could be implications for public health. The current study examines relationships between child and parent reports of screen time use and various important developmental outcome variables in a large and diverse nationwide sample of 9- and 10- year old children collected by the Adolescent Brain Cognitive Development (ABCD) Study [25]. By design, assessing the children for the first time at 9- and 10- years of age allows us to observe adolescent behavioral and psychological relationships at an early stage and prior to the onset of substance experimentation and use, while subsequent waves of assessment will allow us to observe how these relationships change with adolescent development and are modified by experiences including substance experimentation and use. Of additional importance, few studies have examined screen time use at such a young age. The majority of the current literature focuses on mid-to-late adolescence, rather than very early adolescence; the current study's focus on 9- and 10- year old children fills that gap in the literature.

Specific dependent variables of interest are anxiety and depression symptoms, composite internalizing problems, composite externalizing problems, attention problems, attention-deficit hyperactivity disorder symptoms, academic performance, sleep patterns, and peer relationships. In examining these associations, we control for socio-economic status as well as race/ethnicity, as those factors directly impact access to screens and electronic media. Given the previous findings on screen time associations, we ask: in 9- and 10- year old children, what relationships exist between screen time and mental health, behavioral health, academic success, and peer relationships? We hypothesized that total screen time would be 1) positively associated with depression and anxiety symptoms as well as behavioral problems including ADHD, 2) negatively associated with academic performance and sleep quantity and quality, and 3) positively associated with quantity and quality of peer relationships. Our study is unique in its ability to allow us to determine the magnitude of these associations, their importance, and potential adverse impacts of increased screen time in a novel and very large, diverse national sample of 9- to 10- year old children. Our findings lay groundwork for future analyses on the longitudinal ABCD Study sample.

## Materials and methods

### Data

All data were from the existing Adolescent Brain Cognitive Development (ABCD) dataset; the ABCD Study is the largest long-term study of brain development and child and adolescent health in the United States [26]. The ABCD dataset was chosen for analysis because it draws from 21 research sites across the country, which in total recruited 11,875 children, ages 9 to 10 years, primarily from local schools, or, in the case of an embedded twin sample at four sites,

from birth records. Schools in the study were selected in part based on demographic makeup, ensuring the inclusion of all demographic groups. The baseline data collection was completed in October 2018 and subsequent follow-up assessments will occur annually for 10 years, including brain imaging every two years. This resulted in a quasi-representative, longitudinal dataset consisting of both child self-report and parent reported measures of behavioral and psychological characteristics, physical wellness, cognitive function, and environmental factors, and structural and functional brain imaging; biomarkers including DNA for genetic assays were also collected. For parent-report variables, only one parent completed questionnaires, most often the mother. Of note, the entire ABCD sample consists of more males (52.1%) than females (47.8%).

## Participants

Participants were drawn from the 11,875 children enrolled in the Adolescent Brain Cognitive Development (ABCD) Study. We utilized the entire existing dataset because the large sample size, which was determined by the design of the ABCD Study, ensures sufficient power to detect even small associations. Both written and verbal consent were collected from all parents/guardians and from all children. All procedures were approved by a central institutional review board and comply with the World Medical Association Declaration of Helsinki. The University of California San Diego institutional review board has indicated that analyses using the publicly released, anonymized ABCD data are not human subjects research and do not require their own approval. The data used in this study are owned by the National Institute of Mental Health Data Archive; qualified researchers can request access to ABCD shared data at https://nda.nih.gov/abcd/request-access. The ABCD data used in this paper came from NIMH Data Archive Digital Object Identifier 10.15154/1519271.

The sample was 52% White, 20.3% Hispanic, 15% Black, 2.1% Asian, and 10.5% Other or Prefer Not to Respond. The sample tended toward upper-middle SES, with 3.6% reporting annual family income <$5000; 3.6% reporting $5000–$11,999; 2.3% reported $12,000–$15,999; 4.4% reported $16,000–$24,999; 5.5% reporting $25,000–$34,999; 7.9% reporting $35,000–$49,999; 12.6% reporting $50,000–$74,999; 13.2% reporting $75,000–$99,999; 27.9% reporting $100,000–$199,999; 10.5% reporting $200,000+; with 4.3% refuse to answer and 4.2% don't know.

We divided the current study into two parts to better assess fundamental differences in weekday and weekend screen time use. There exists a significant difference in time spent on screens, t(11,723) = -53.34, p < .001, during a weekday (M = 3.39, SD = 2.94) and time spent on screens during the weekend (M = 4.57, SD = 3.54). There was also a significant difference in parent reports of their child's screen time, t(11,748) = -61.63, p < .001, between weekdays (M = 2.55, SD = 2.59) and weekends (M = 3.99, SD = 2.66). Additionally, there existed significant differences between weekday and weekend screen usage type (e.g., TV/movies, online videos, gaming, etc.) for every usage type examined (see S1 Table). These significant differences suggest that weekday and weekend screen time and screen time use type differ, and therefore, should be examined separately.

**Part 1.** Part 1 examines weekday screen time use only, which is defined as media use per individual day of the week (Monday through Friday). Of the original 11,875 participants, 41 were excluded for implausible self-reported weekday screen time use of 18 hours or more, 69 were excluded for implausible parent report of weekday screen time use of 18 hours or more, and 38 were excluded for missing data. Thus, the current analyses were conducted on a sample of 11,727 children ages 9 to 10.92 years (M = 9.91 years, SD = .62 years). The sample remained representative of the ABCD Study, with 6111 males (52.1%), 5613 females (47.9%), and 3 individuals who chose not to disclose their sex.

**Part 2.** Part 2 examines weekend screen time use only, which is defined as media use per individual day of the weekend (Saturday and Sunday). We returned to the original 11,875 participants and followed similar exclusion principles as in Part 1. First, 115 participants were excluded for self-reported weekend screen time use of 18 hours or more per day. Next, 49 were excluded for parent report of weekend screen time use of 18 hours or more per day, and 37 were excluded for missing data. The current analyses were conducted on a sample of 11,672 children ages 9 to 10.92 years, ($M = 9.91$ years, $SD = 0.62$ years). The sample remained representative of the ABCD Study, with 6071 males (52%), 5598 females (48%), and 3 individuals who chose not to disclose their sex.

## Measures

**Screen time.** A 14-question Screen Time Questionnaire (STQ) was completed by the children, providing self-report measures of screen time use, divided by weekdays and weekends. The questionnaire asks how many hours per weekday/weekend day the child uses screens for different types of media, with responses ranging from "0 h" (0) to "4 + h" (4). The STQ divides screen time use among six different forms of recreational (not for schoolwork) media use: television shows and movies, videos, video games, texting, social media, and video chat. The total amount of time spent on screens on an individual weekday or weekend day is a composite across all six forms of media types. The questionnaire also asks children to report the frequency with which they engage in mature video gaming and R-rated movie viewing (0 = never, 3 = all the time). The child's parent/guardian also completed a shorter STQ, which asked about the child's total screen time on individual weekdays/weekend days in both hours and minutes (e.g., a parent could report that their child spends 2 hours and 30 minutes on a screen). Screen time in hours was used in this analysis (e.g. 2 hours and 30 minutes = 2.5 hours).

**Depression.** The parent/guardian of the child participant completed the 112-item Child Behavior Checklist (CBCL), which asks parents about various psychiatric symptoms and behaviors the child shows [27]. Participants' depression symptoms were evaluated on a sub-scale containing 13 statements, to which parents of the participant reported on a scale from 0 (not true) to 2 (very true/often true) in response to statements about their child, which included items concerning withdrawal and depressed mood. The resulting CBCL derived T-score for depression was used for analysis; subsequent CBCL measures of interest were also analyzed via T-scores.

**Anxiety.** Participant anxiety was evaluated by the CBCL on a subscale of 9 statements. Participants' parents reported on a scale from 0 (not true) to 2 (very true/often true) in response to statements about their child that concerned anxious behavior.

**Internalizing problems.** Internalizing problems is a composite score on the CBCL, calculated by summing the total depression, anxiety, and somatic scores for the participant.

**Externalizing problems.** Externalizing behavioral problems is a composite score on the CBCL, calculated by summing the participant's total rule-breaking behavior and aggressive behavior scores.

**Oppositional defiant disorder (ODD).** The ODD subscale is one of six DSM-oriented scales within the CBCL that is consistent with DSM-5 diagnosis. Parents reported on a scale from 0 (not true) to 2 (very true/often true) in response to statements about their child's behavior in accordance with DSM-5 criterion for oppositional defiant problems.

**Conduct disorder.** Conduct disorder is another of six DSM-oriented scales within the CBCL that is consistent with DSM-5 diagnosis. Parents reported on a scale from 0 (not true) to 2 (very true/often true) in response to statements about their child's behavior in accordance with DSM-5 criterion for conduct problems.

**Attention problems.**   Participant attention problems were measured by the CBCL on a subscale, to which parents reported on a scale from 0 (not true) to 2 (very true/often true) in response to statements about their child concerning attentional problems.

**Attention-deficit hyperactivity disorder (ADHD).**   ADHD is one of six DSM-oriented scales within the CBCL that is consistent with DSM-5 diagnosis. Parents reported on a scale from 0 (not true) to 2 (very true/often true) in response to statements about their child's attention behavior in accordance with DSM-5 criterion for ADHD.

**Academic performance.**   Academic performance was measured by participants' grades in school, which were reported by their parents in response to the question, "What kind of grades does your child get on average?" Parents reported if their child earned As, Bs, Cs, Ds, or Fs (1 = As, 5 = Fs) or they selected N/A (-1) if not applicable. To conduct analysis, the variable for grades were re-coded so that higher codes corresponded with better grades (1 = Fs, 5 = As) and N/As were set to missing.

**Sleep quantity.**   Participant sleep habits were partially measured by the quantity of sleep the participant typically gets per night. The average number of hours of sleep per night were reported by participants' parents in response to the question, "How many hours of sleep does your child get on most nights?" Parents reported if their child typically sleeps 9–11 hours, 8–9 hours, 7–8 hours, 5–7 hours, or less than 5 hours (1 = 9–11 hours, 5 = less than 5 hours). To conduct analysis, the variable for amount of sleep was re-coded so that higher codes corresponded with more sleep (1 = less than 5 hours, 5 = 9–11 hours).

**Sleep quality.**   Participant sleep habits were also measured by the quality of sleep the participant typically has. Participants' general sleep quality was reported by their parents in response to a series of questions that produced scores indicative of six different sleep disorders: disorders of arousal, disorders of initiating and maintaining sleep, disorders of excessive somnolence, sleep breathing disorders, sleep hyperhidrosis, and sleep-wake transition disorders. Scores across disorders were summed into a total sleep disorder score, with a higher score meaning higher incidents of sleep disorders, and thus, poorer quality sleep.

**Number of close friends who are boys.**   The quantity and quality of peer relationships was measured by the number of close friends a participant has. By specifically examining the number of *close* friends, rather than merely the total number of friends, we can assume that these friendships are of quality to the participant. The questionnaire divided friendships by sex of the friend; first, participants were asked to report how many close friends *who are boys* they have.

**Number of close friends who are girls.**   Participants were also asked to report how many *close* friends *who are girls* they have. The correlation between the self-report number of close friends who are boys and number of close friends who are girls was weak across both Part 1 and Part 2, so these outcomes were analyzed separately.

**Combined family income.**   Parents/guardians reported the total combined family income before taxes for the previous 12 months. Income responses were coded as 1 = < $5,000; 2 = $5,000 - $11,999; 3 = $12,000 - $15,999; 4 = $16,000 - $24,999; 5 = $25,000 - $34,999; 6 = $35,000 - $49,999; 7 = $50,000 - $74,999; 8 = $75,000 - $99,999; 9 = $100,000 - $199,999; and 10 = $200,000+. Responses "refuse to answer" and "don't know" were set to missing for analysis.

**Race/Ethnicity.**   Child race/ethnicity was obtained via both parent and self-report. Responses were coded as 1 = White; 2 = Black; 3 = Hispanic; 4 = Asian; 5 = Other. Distributions for each study are reported above.

## Statistical analysis

All statistical analyses were conducted with IBM SPSS Statistics Version 26. Bivariate Pearson correlations between each of the variables were computed to evaluate the interrelationships

between all variables, including the different measures of screen time. Correlations were calculated separately for each sex. We conducted a combined regression (across sex), coding sex as a dummy variable to investigate—via interaction test—whether the effect of screen time on our outcome variables depended on sex. Sex was dummy coded with females = 0 and males = 1, making "females" the base category for comparison. Multiple linear regressions were then run, separately by sex, with the different measures for screen time as the predictor and each outcome variable as the dependent variable. All regressions controlled for both SES and race/ethnicity. The analyses conducted rely on the normal distribution assumption; the independent variables (screen time) and depended variables are only approximately normally distributed and thus p-values are necessarily subject to some imprecision. Thirteen primary regressions were conducted; to account for multiple testing, the Bonferroni corrected significance level was .004 for our primary test: the interaction test and the investigation of relationships between weekend/weekday screen time total and mental health, behavioral health, academic performance, sleep quality and quantity, and peer relationships. The significance level for all secondary tests was .05. We conducted analyses separately by sex because there existed significant sex differences in total weekday screen time, $t(11,831) = 10.22$, p < .001, with males ($M = 3.74$, $SD = 3.17$) spending more weekday time on screens than females ($M = 3.16$, $SD = 2.99$); in total weekend screen time, $t(11,829) = 13.54$, p < .001, with males ($M = 5.05$, $SD = 3.68$) spending more weekend time on screens than females ($M = 4.16$, $SD = 3.53$); as well as each outcome measure. The sex differences suggested that males and females differed in both independent and dependent variables, and therefore, should be examined separately. Subsequent analyses were conducted separately by sex. Table 1 provides sex differences, separated by Parts 1 and 2.

**Part 1.** Only weekday screen time measures were included. Sex differences between each screen time type and each outcome measure were examined with a two-tailed independent samples t-test, alpha level .05, and are displayed in the top section of Table 1.

**Part 2.** Only weekend screen time measures were included. Sex differences between each screen time type and each outcome measure were examined as in Part 1 and are displayed in the lower section of Table 1.

## Results

### Part 1

Correlations between all variables, separately by sex, are shown in S2 Table. While the majority of correlations are significant, most are weak or moderate in strength. Measures that one would expect to be correlated are (e.g., the correlation between attention problems and ADHD is strong for both sexes). The data do not demonstrate multicollinearity, as seen in S3 Table.

Examination of whether the effect of weekday screen time on our outcome variables of interest depended on sex yielded interesting results. The vast majority of interactions were not significant; however, both main effects of weekday screen time and sex were often significant at Bonferroni corrected alpha .004, as seen in Table 2. Our primary interest was examination of the effects of screen time and sex on our dependent variables; however, we also report results for race/ethnicity and SES for the sake of completeness. The main effect of SES was also often significant.

Because there is a demonstrated significant main effect of sex, it was important to also run Multiple regression separately by sex to more closely examine sex differences. When running Multiple Regression separately by sex, the majority of regressions of outcome variables on total weekday screen time were significant using a Bonferroni corrected p-value of less than .004, and were in line with our hypotheses, controlling for SES and race/ethnicity, as shown in Table 3. Effect sizes for each of these tests are small and also shown in Table 3.

**Table 1. Participant sex differences on screen time measures and outcome variables for Part 1 and Part 2.**

|  | Males Mean (*SD*) | Females Mean (*SD*) | t statistic | p-value |
|---|---|---|---|---|
| **Part 1** | **(N = 6111)** | **(N = 5613)** |  |  |
| Total Screen Time | 3.67 (*3.01*) | 3.09 (*2.82*) | 10.72 | < .001* |
| Parent-Report Total | 2.56 (*2.29*) | 2.32 (*2.06*) | 5.91 | < .001* |
| TV and Movies | 1.12 (*1.10*) | 1.10 (*1.09*) | 1.05 | .294 |
| Videos | 0.96 (*1.18*) | 0.83 (*1.11*) | 6.50 | < .001* |
| Video Chat | 0.15 (*0.43*) | 0.19 (*0.46*) | -4.81 | < .001* |
| Texting | 0.17 (*0.47*) | 0.24 (*0.55*) | -7.39 | < .001* |
| Social Media | 0.08 (*0.35*) | 0.12 (*0.41*) | -5.21 | < .001* |
| Video Games | 1.19 (*1.24*) | 0.62 (*0.91*) | 28.14 | < .001* |
| Mature Games | 0.82 (*0.98*) | 0.28 (*0.60*) | 35.72 | < .001* |
| R-rated Movies | 0.43 (*0.67*) | 0.32 (*0.59*) | 9.49 | < .001* |
| Depression | 54.23 (*6.31*) | 52.72 (*5.05*) | 14.22 | < .001* |
| Anxiety | 53.79 (*6.15*) | 53.13 (*5.73*) | 6.01 | < .001* |
| Internalizing | 49.35 (*10.67*) | 47.44 (*10.50*) | 9.75 | < .001* |
| Externalizing | 46.47 (*10.65*) | 44.86 (*9.85*) | 8.49 | < .001* |
| Oppositional defiance | 53.95 (*5.81*) | 52.93 (*4.86*) | 10.24 | < .001* |
| Conduct disorder | 53.28 (*5.70*) | 52.71 (*5.28*) | 5.65 | < .001* |
| Attention problems | 54.24 (*6.49*) | 53.51 (*5.73*) | 6.42 | < .001* |
| ADHD | 53.63 (*6.00*) | 52.75 (*5.14*) | 8.50 | < .001* |
| Academic performance | 4.23 (*0.84*) | 4.41 (*0.74*) | -11.42 | < .001* |
| Sleep quantity | 4.28 (*0.82*) | 4.30 (*0.80*) | -1.01 | .313 |
| Sleep quality | 36.78 (*8.46*) | 36.22 (*7.92*) | 3.73 | < .001* |
| Num. of close m. friends | 4.45 (*6.92*) | 1.30 (*2.49*) | 33.14 | < .001* |
| Num. of close f. friends | 1.69 (*4.85*) | 5.13 (*7.37*) | -30.04 | < .001* |
| **Part 2** | **(N = 6071)** | **(N = 5598)** |  |  |
| Total Screen Time | 4.88 (*3.32*) | 4.00 (*3.16*) | 14.68 | < .001* |
| Parent-Report Total | 4.09 (*2.47*) | 3.70 (*2.31*) | 8.69 | .003* |
| TV and Movies | 1.62 (*1.27*) | 1.61 (*1.25*) | 0.45 | .084 |
| Videos | 1.22 (*1.34*) | 1.01 (*1.25*) | 8.90 | < .001* |
| Video Chat | 0.16 (*0.48*) | 0.22 (*0.54*) | -6.44 | < .001* |
| Texting | 0.18 (*0.48*) | 0.26 (*0.59*) | -8.40 | < .001* |
| Social Media | 0.08 (*0.35*) | 0.14 (*0.49*) | -7.51 | < .001* |
| Video Games | 1.62 (*1.37*) | 0.75 (*1.03*) | 38.14 | < .001* |
| Mature Games | 0.81 (*0.97*) | 0.28 (*0.60*) | 35.43 | < .001* |
| R-rated Movies | 0.43 (*0.66*) | 0.32 (*0.59*) | 9.21 | < .001* |
| Depression | 54.22 (*6.29*) | 52.72 (*5.03*) | 14.20 | < .001* |
| Anxiety | 53.78 (*6.49*) | 53.12 (*5.71*) | 6.00 | < .001* |
| Internalizing | 49.32 (*10.66*) | 47.45 (*10.49*) | 9.54 | .008* |
| Externalizing | 46.41 (*10.63*) | 44.84 (*9.85*) | 8.26 | < .001* |
| Oppositional defiance | 53.91 (*5.77*) | 52.92 (*4.85*) | 9.95 | < .001* |
| Conduct disorder | 53.26 (*5.66*) | 52.70 (*5.29*) | 5.47 | < .001* |
| Attention problems | 54.21 (*6.49*) | 53.50 (*5.74*) | 6.23 | < .001* |
| ADHD | 53.60 (*5.98*) | 52.75 (*5.14*) | 8.24 | < .001* |
| Academic performance | 4.24 (*0.83*) | 4.41 (*0.74*) | -11.18 | < .001* |
| Sleep quantity | 4.29 (*0.82*) | 4.30 (*0.80*) | -1.04 | .058 |
| Sleep quality | 36.73 (*8.33*) | 36.21 (*7.91*) | 3.44 | < .001* |
| Num. of close m. friends | 4.52 (*6.81*) | 1.29 (*2.46*) | 33.56 | < .001* |

(*Continued*)

**Table 1.** (Continued)

|  | **Males Mean (*SD*)** | **Females Mean (*SD*)** | **t statistic** | **p-value** |
|---|---|---|---|---|
| Num. of close f. friends | 1.67 (*4.68*) | 5.11 (*7.44*) | -30.16 | < .001* |

*Note*. Starred significance at .05. Screen time measure means given in hours. ADHD = attention/deficit hyperactivity disorder; Num. of Close M. Friends = number of close friends who are male; Num. of Close F. Friends = number of close friends who are female.

As seen in Table 3, for males, total weekday screen time is significantly associated with internalizing problems, externalizing problems, ODD, conduct disorder, attention problems, ADHD, academic performance, sleep quantity, sleep quality, the number of close friends who are male, and the number of close friends who are female. However, weekday total screen time is not significantly associated with either depression or anxiety.

Additionally seen in Table 3, for females, total weekday screen time is significantly associated with externalizing problems, ODD, conduct disorder, attention problems, ADHD, academic performance, sleep quantity, sleep quality, the number of close friends who are male, and the number of close friends who are female. Weekday total screen time is not significantly associated with depression, anxiety, or internalizing problems.

We were secondarily interested in the relationships between our outcome measures and differing forms of weekday screen time use (e.g., social media versus video viewing). For males, relationships between differing types of screen time and depression, anxiety, and internalizing problems are not significant at alpha .05. For all other outcome measures, the majority of associations between the various types of screen time and that outcome are significant. Of the types of screen time, video chat and texting have the least reliable predictive power and are only significant for some outcomes. Parent report of total screen time is significant for about half of outcome variables.

Similarly for females, relationships between differing types of screen time and depression, anxiety, and internalizing problems are not significant. For all other outcome measures, the majority of associations between the various types of screen time and that outcome are significant. Of the types of screen time, video chat has the least reliable predictive power, and is significant in about half of outcome measures. Parent report of total screen time significantly predicted the outcome measures for the majority of tests, apart from the number of close friends who are boys and the number of close friends who are girls.

The comprehensive results of our statistical analyses for Part 1, including effect sizes, are displayed in S6–S18 Tables.

## Part 2

Correlations between all variables, separately by sex, are shown in S4 Table. As in Part 1, the majority of correlations are significant, are weak or moderate in strength, and measures that one would expect to be correlated are. The data do not demonstrate multicollinearity, as seen in S5 Table.

Once again, examination of whether the effect of weekend screen time on our outcome variables of interest depended on sex yielded varied results. The vast majority of interactions were not significant; however, as in Part 1, both main effects of weekday screen time and sex were often significant at Bonferroni corrected alpha .004, as seen in Table 4. As in Part 1, our primary interest was examination of the effects of screen time and sex on our dependent variables; however, we also report results for race/ethnicity and SES for the sake of completeness. The main effect of SES was also often significant.

**Table 2. Examination of main effects of weekday screen time and sex, as well as the interaction between them, on outcome variables, controlling for SES and race/ethnicity.**

| | Standardized Beta | t statistic | p-value | Standard Error | Partial Correlation |
|---|---|---|---|---|---|
| **Depression** | | | | | |
| Main effect of ST | 0.018 | 1.24 | .214 | .029 | .012 |
| Main effect of sex | 0.112 | 7.78 | < .001* | .166 | .075 |
| Main effect of R/E | -0.007 | -0.72 | .472 | .042 | -.01 |
| Main effect of SES | -0.141 | -14.01 | < .001* | .024 | -.13 |
| Interaction sex*ST | 0.028 | 1.48 | .139 | .038 | .014 |
| **Anxiety** | | | | | |
| Main effect of ST | -0.002 | -0.16 | .877 | .030 | -.001 |
| Main effect of sex | 0.036 | 2.42 | .016 | .176 | .023 |
| Main effect of R/E | -0.018 | -1.86 | .063 | .044 | -.018 |
| Main effect of SES | -0.045 | -4.36 | < .001* | .025 | -.042 |
| Interaction sex*ST | 0.031 | 1.66 | .097 | .040 | .016 |
| **Internalizing** | | | | | |
| Main effect of ST | 0.007 | 0.45 | .655 | .054 | .004 |
| Main effect of sex | 0.065 | 4.44 | < .001* | .310 | .043 |
| Main effect of R/E | -0.011 | -1.09 | .278 | .078 | -.010 |
| Main effect of SES | -0.077 | -7.49 | < .001* | .045 | -.072 |
| Interaction sex*ST | 0.036 | 1.91 | .056 | .071 | .018 |
| **Externalizing** | | | | | |
| Main effect of ST | 0.083 | 5.69 | < .001* | .051 | .055 |
| Main effect of sex | 0.069 | 4.73 | < .001* | .297 | .046 |
| Main effect of R/E | -0.014 | -1.41 | .159 | .075 | -.014 |
| Main effect of SES | -0.142 | -14.03 | < .001* | .043 | -.134 |
| Interaction sex*ST | 0.001 | 0.04 | .967 | .068 | .000 |
| **Oppositional defiance** | | | | | |
| Main effect of ST | 0.066 | 4.47 | < .001* | .027 | .043 |
| Main effect of sex | 0.077 | 5.31 | < .001* | .157 | .051 |
| Main effect of R/E | -0.021 | -2.19 | .029 | .040 | -.021 |
| Main effect of SES | -0.090 | -8.86 | < .001* | .023 | -.085 |
| Interaction sex*ST | 0.014 | 0.76 | .450 | .036 | .007 |
| **Conduct disorder** | | | | | |
| Main effect of ST | 0.096 | 6.65 | < .001* | .027 | .064 |
| Main effect of sex | 0.037 | 2.60 | .009 | .157 | .025 |
| Main effect of R/E | -0.006 | -0.62 | .539 | .040 | -.006 |
| Main effect of SES | -0.165 | -16.43 | < .001* | .023 | -.157 |
| Interaction sex*ST | 0.008 | 0.43 | .664 | .036 | .004 |
| **Attention problems** | | | | | |
| Main effect of ST | 0.074 | 5.07 | < .001* | .031 | .049 |
| Main effect of sex | 0.044 | 3.02 | .003* | .178 | .029 |
| Main effect of R/E | 0.011 | 1.12 | .261 | .045 | .011 |
| Main effect of SES | -0.095 | -9.36 | < .001* | .026 | -.090 |
| Interaction sex*ST | 0.013 | 0.71 | .480 | .041 | .007 |
| **ADHD** | | | | | |
| Main effect of ST | 0.088 | 6.04 | < .001* | .028 | .058 |
| Main effect of sex | 0.056 | 3.85 | < .001* | .163 | .037 |
| Main effect of R/E | 0.010 | 1.02 | .310 | .041 | .010 |

*(Continued)*

**Table 2.** (Continued)

|  | Standardized Beta | t statistic | p-value | Standard Error | Partial Correlation |
|---|---|---|---|---|---|
| Main effect of SES | -0.089 | -8.80 | < .001* | .024 | -.085 |
| Interaction sex*ST | 0.022 | 1.15 | .250 | .037 | .011 |
| **Academic performance** | | | | | |
| Main effect of ST | -0.107 | -7.37 | < .001* | .004 | -.074 |
| Main effect of sex | -0.083 | -5.69 | < .001* | .023 | -.057 |
| Main effect of R/E | -0.041 | -4.24 | < .001* | .006 | -.043 |
| Main effect of SES | 0.253 | 25.04 | < .001* | .003 | .244 |
| Interaction sex*ST | -0.010 | -0.54 | .593 | .005 | -.005 |
| **Sleep quantity** | | | | | |
| Main effect of ST | -0.149 | -10.69 | < .001* | .004 | -.103 |
| Main effect of sex | 0.011 | 0.80 | .426 | .022 | .007 |
| Main effect of R/E | -0.065 | -7.03 | < .001* | .006 | -.068 |
| Main effect of SES | 0.246 | 25.53 | < .001* | .003 | .239 |
| Interaction sex*ST | -0.018 | -1.00 | .316 | .005 | -.009 |
| **Sleep quality** | | | | | |
| Main effect of ST | 0.054 | 3.70 | < .001* | .041 | .036 |
| Main effect of sex | 0.008 | 0.58 | .565 | .236 | .006 |
| Main effect of R/E | 0.012 | 1.19 | .235 | .060 | .011 |
| Main effect of SES | -0.099 | -9.74 | < .001* | .034 | -.094 |
| Interaction sex*ST | 0.031 | 1.66 | .096 | .054 | .016 |
| **Num. of close m. friends** | | | | | |
| Main effect of ST | 0.029 | 2.06 | .039 | .026 | .020 |
| Main effect of sex | 0.247 | 17.60 | < .001* | .150 | .168 |
| Main effect of R/E | -0.013 | -1.34 | .182 | .038 | -.013 |
| Main effect of SES | 0.012 | 1.26 | .209 | .022 | .012 |
| Interaction sex*ST | 0.077 | 4.26 | < .001* | .034 | .041 |
| **Num. of close f. friends** | | | | | |
| Main effect of ST | 0.067 | 4.67 | < .001* | .031 | .045 |
| Main effect of sex | -0.285 | -20.15 | < .001* | .179 | -.191 |
| Main effect of R/E | -0.017 | -1.79 | .074 | .045 | -.017 |
| Main effect of SES | -0.014 | -1.376 | .169 | .026 | -.013 |
| Interaction sex*ST | 0.001 | 0.08 | .934 | .041 | .001 |

*Note*: Starred regressions are significant at Bonferroni corrected alpha .004. ST = weekday screen time in hours. R/E = race/ethnicity.

Because there is a significant main effect of sex, it was important to also run Multiple regression separately by sex to more closely examine sex differences. When running Multiple Regression separately by sex, the majority of regressions of outcome variables on total weekend screen time are significant using a Bonferroni corrected p-value of less than .004 and are in line with our hypotheses, controlling for SES and race/ethnicity, as shown in Table 5. Effect sizes for each of these tests are small and are also shown in Table 5.

As seen in Table 5, for males, total weekend screen time is significantly associated with all outcome measures: depression, anxiety, internalizing problems, externalizing problems, ODD, conduct disorder, attention problems, ADHD, academic performance, sleep quantity, sleep quality, the number of close friends who are male, and the number of close friends who are female.

**Table 3. Outcome measures regressed on weekday total screen time for Part 1, controlling for SES and race/ethnicity, separated by sex.**

| | Standardized Beta | t statistic | p-value | Standard Error | Partial Correlation |
|---|---|---|---|---|---|
| **Males (N = 6111)** | | | | | |
| Depression | 0.036 | 2.58 | 0.010 | .029 | .035 |
| Anxiety | 0.028 | 1.97 | 0.049 | .029 | .026 |
| Internalizing | 0.043 | 3.06 | .002* | .050 | .041 |
| Externalizing | 0.076 | 5.48 | < .001* | .049 | .073 |
| Oppositional defiance | 0.068 | 4.88 | < .001* | .027 | .065 |
| Conduct disorder | 0.093 | 6.84 | < .001* | .026 | .091 |
| Attention problems | 0.079 | 5.70 | < .001* | .030 | .076 |
| ADHD | 0.102 | 7.34 | < .001* | .028 | .098 |
| Academic performance | -0.107 | -7.70 | < .001* | .004 | -.107 |
| Sleep quantity | -0.171 | -12.97 | < .001* | .004 | -.171 |
| Sleep quality | 0.081 | 5.81 | < .001* | .039 | .078 |
| Num. of close m. friends | 0.092 | 6.58 | < .001* | .031 | .088 |
| Num. of close f. friends | 0.079 | 5.69 | < .001* | .022 | .076 |
| **Females (N = 5613)** | | | | | |
| Depression | 0.029 | 2.03 | 0.042 | .026 | .028 |
| Anxiety | 0.000 | 0.002 | 0.999 | .030 | .000 |
| Internalizing | 0.008 | 0.56 | 0.579 | .054 | .008 |
| Externalizing | 0.091 | 6.35 | < .001* | .050 | .088 |
| Oppositional defiance | 0.079 | 5.51 | < .001* | .025 | .077 |
| Conduct disorder | 0.107 | 7.50 | < .001* | .027 | .104 |
| Attention problems | 0.083 | 5.81 | < .001* | .029 | .081 |
| ADHD | 0.097 | 6.80 | < .001* | .026 | .094 |
| Academic performance | -0.118 | -8.27 | < .001* | .004 | -.119 |
| Sleep quantity | -0.143 | -10.56 | < .001* | .004 | -.146 |
| Sleep quality | 0.060 | 4.16 | < .001* | .041 | .058 |
| Num. of close m. friends | 0.053 | 3.66 | < .001* | .013 | .051 |
| Num. of close f. friends | 0.065 | 4.50 | < .001* | .038 | .063 |

*Note.* Starred regressions are significant at Bonferroni corrected alpha .004. ADHD = attention/deficit hyperactivity disorder; Num. of close m. friends = number of close friends who are male; Num. of close f. friends = number of close friends who are female.

As seen in Table 5, for females, total weekend screen time is significantly associated with externalizing problems, ODD, conduct disorder, attention problems, ADHD, academic performance, sleep quantity, sleep quality, the number of close friends who are male, and the number of close friends who are female. Weekend total screen time is not significantly associated with depression, anxiety, or internalizing problems.

We were secondarily interested in the relationships between our outcome measures and differing types of weekend screen time use. For males, although total weekend screen time demonstrated a significant relationship to every outcome variable at alpha .05, the overall relationships between differing types of screen time and depression, anxiety, and internalizing problems are not significant. For all other outcome measures, the majority of associations between the various types of screen time and that outcome are significant. Of the types of screen time, video chat and texting have the least reliable predictive power and are only significantly associated with some outcomes. Parent report of total weekend screen time is significant for all outcomes apart from the number of close friends who are boys and the number of close friends who are girls.

**Table 4. Examination of main effects of weekend screen time and sex, as well as the interaction between them, on outcome variables, controlling for SES and race/ethnicity.**

| | Standardized Beta | t statistic | p-value | Standard Error | Partial Correlation |
|---|---|---|---|---|---|
| **Depression** | | | | | |
| Main effect of ST | 0.014 | 0.98 | .327 | .025 | .009 |
| Main effect of sex | 0.102 | 6.35 | < .001* | .183 | .061 |
| Main effect of R/E | -0.005 | -0.55 | .584 | .042 | -.005 |
| Main effect of SES | -0.140 | -14.04 | < .001* | .024 | -.135 |
| Interaction sex*ST | 0.041 | 2.02 | .043 | .034 | .020 |
| **Anxiety** | | | | | |
| Main effect of ST | -0.001 | -0.05 | .959 | .027 | .000 |
| Main effect of sex | 0.015 | 0.92 | .358 | .194 | .009 |
| Main effect of R/E | -0.017 | -1.75 | .080 | .044 | -.017 |
| Main effect of SES | -0.040 | -3.98 | < .001* | .025 | -.039 |
| Interaction sex*ST | 0.057 | 2.80 | .005 | .036 | .027 |
| **Internalizing** | | | | | |
| Main effect of ST | 0.017 | 1.14 | .253 | .047 | .011 |
| Main effect of sex | 0.054 | 3.32 | .001* | .343 | .032 |
| Main effect of R/E | -0.010 | -0.97 | .334 | .078 | -.009 |
| Main effect of SES | -0.072 | -7.14 | < .001* | .044 | -.069 |
| Interaction sex*ST | 0.044 | 2.18 | .030 | .063 | .021 |
| **Externalizing** | | | | | |
| Main effect of ST | 0.096 | 6.70 | < .001* | .045 | .065 |
| Main effect of sex | 0.067 | 4.18 | < .001* | .329 | .040 |
| Main effect of R/E | -0.013 | -1.37 | .172 | .075 | -.013 |
| Main effect of SES | -0.141 | -14.13 | < .001* | .042 | -.136 |
| Interaction sex*ST | -0.005 | -0.26 | .796 | .061 | -.003 |
| **Oppositional defiance** | | | | | |
| Main effect of ST | 0.074 | 5.10 | < .001* | .024 | .049 |
| Main effect of sex | 0.069 | 4.26 | < .001* | .173 | .041 |
| Main effect of R/E | -0.022 | -2.24 | .025 | .040 | -.022 |
| Main effect of SES | -0.089 | -8.83 | < .001* | .022 | -.085 |
| Interaction sex*ST | 0.017 | 0.84 | .402 | .032 | .008 |
| **Conduct disorder** | | | | | |
| Main effect of ST | 0.104 | 7.25 | < .001* | .024 | .070 |
| Main effect of sex | 0.038 | 2.37 | .018 | .174 | .023 |
| Main effect of R/E | -0.004 | -0.45 | .651 | .040 | -.004 |
| Main effect of SES | -0.166 | -17.72 | < .001* | .022 | -.160 |
| Interaction sex*ST | 0.000 | -0.01 | .989 | .032 | .000 |
| **Attention problems** | | | | | |
| Main effect of ST | 0.103 | 7.13 | < .001* | .027 | .069 |
| Main effect of sex | 0.035 | 2.18 | .029 | .197 | .021 |
| Main effect of R/E | 0.009 | 0.89 | .376 | .045 | .009 |
| Main effect of SES | -0.092 | -9.15 | < .001* | .025 | -.088 |
| Interaction sex*ST | 0.015 | 0.73 | .468 | .036 | .007 |
| **ADHD** | | | | | |
| Main effect of ST | 0.121 | 8.43 | < .001* | .025 | .081 |
| Main effect of sex | 0.050 | 3.09 | .002* | .180 | .030 |
| Main effect of R/E | 0.007 | 0.68 | .495 | .041 | .007 |

*(Continued)*

**Table 4.** (Continued)

|  | Standardized Beta | t statistic | p-value | Standard Error | Partial Correlation |
|---|---|---|---|---|---|
| Main effect of SES | -0.086 | -8.63 | < .001* | .023 | -.083 |
| Interaction sex*ST | 0.015 | 0.77 | .444 | .033 | .007 |
| **Academic performance** | | | | | |
| Main effect of ST | -0.076 | -5.30 | < .001* | .003 | -.053 |
| Main effect of sex | -0.070 | -4.33 | < .001* | .025 | -.044 |
| Main effect of R/E | -0.043 | -4.38 | < .001* | .006 | -.044 |
| Main effect of SES | 0.264 | 26.32 | < .001* | .003 | .256 |
| Interaction sex*ST | -0.026 | -1.30 | .194 | .005 | -.013 |
| **Sleep quantity** | | | | | |
| Main effect of ST | -0.139 | -10.08 | < .001* | .003 | -.097 |
| Main effect of sex | 0.006 | 0.40 | .693 | .025 | .004 |
| Main effect of R/E | -0.066 | -7.08 | < .001* | .006 | -.068 |
| Main effect of SES | 0.257 | 26.82 | < .001* | .003 | .251 |
| Interaction sex*ST | -0.006 | -0.33 | .739 | .005 | -.003 |
| **Sleep quality** | | | | | |
| Main effect of ST | 0.072 | 4.98 | < .001* | .036 | .048 |
| Main effect of sex | 0.001 | 0.09 | .928 | .261 | .001 |
| Main effect of R/E | 0.010 | 1.04 | .300 | .059 | .010 |
| Main effect of SES | -0.095 | -9.47 | < .001* | .034 | -.091 |
| Interaction sex*ST | 0.031 | 1.53 | .127 | .048 | .015 |
| **Num. of close m. friends** | | | | | |
| Main effect of ST | 0.043 | 3.06 | .002* | .022 | .030 |
| Main effect of sex | 0.249 | 16.01 | < .001* | .163 | .153 |
| Main effect of R/E | -0.014 | -1.45 | .146 | .037 | -.014 |
| Main effect of SES | 0.010 | 1.042 | .297 | .021 | .010 |
| Interaction sex*ST | 0.070 | 3.60 | < .001* | .030 | .035 |
| **Num. of close f. friends** | | | | | |
| Main effect of ST | 0.067 | 4.74 | < .001* | .027 | .046 |
| Main effect of sex | -0.279 | -17.75 | < .001* | .198 | -.170 |
| Main effect of R/E | -0.016 | -1.64 | .101 | .045 | -.016 |
| Main effect of SES | -0.017 | -1.73 | .083 | .026 | -.017 |
| Interaction sex*ST | -0.014 | -0.71 | .477 | .036 | -.007 |

*Note*: Starred regressions are significant at Bonferroni corrected alpha .004. ST = weekend screen time in hours. R/E = race/ethnicity.

For females, relationships between differing types of weekend screen time and depression, anxiety, and internalizing problems are not significant. For all other outcome measures, the majority of associations between the various types of screen time and that outcome are significant. Of the types of screen time, both video chat and texting have the least reliable predictive power and are significant in only about half of outcome measures. Parent report of total weekend screen time is significantly related to all outcome variables.

The comprehensive results of our statistical analyses for Part 2, including effect sizes, are displayed in S19–S31 Tables.

## Discussion

These results have important implications. The lack of consistently significant interactions between screen time and sex—but often significant main effects for both screen time and

**Table 5. Outcome measures regressed on weekend total screen time for Part 2, controlling for SES and race/ethnicity, separated by sex.**

| | Standardized Beta | t statistic | p-value | Standard Error | Partial Correlation |
|---|---|---|---|---|---|
| **Males (N = 6071)** | | | | | |
| Depression | 0.044 | 3.22 | .001* | .007 | .043 |
| Anxiety | 0.052 | 3.75 | < .001* | .026 | .050 |
| Internalizing | 0.058 | 4.24 | < .001* | .044 | .057 |
| Externalizing | 0.084 | 6.21 | < .001* | .044 | .083 |
| Oppositional defiance | 0.079 | 5.77 | < .001* | .024 | .077 |
| Conduct disorder | 0.094 | 6.96 | < .001* | .023 | .093 |
| Attention problems | 0.108 | 7.88 | < .001* | .027 | .105 |
| ADHD | 0.126 | 9.25 | < .001* | .025 | .123 |
| Academic performance | -0.092 | -6.70 | < .001* | .003 | -.094 |
| Sleep quantity | -0.145 | -11.14 | < .001* | .003 | -.148 |
| Sleep quality | 0.096 | 7.03 | < .001* | .034 | .094 |
| Num. of close m. friends | 0.091 | 6.57 | < .001* | .027 | .088 |
| Num. of close f. friends | 0.062 | 4.52 | < .001* | .018 | .061 |
| **Females (N = 5598)** | | | | | |
| Depression | 0.022 | 1.57 | 0.117 | .007 | .022 |
| Anxiety | 0.001 | 0.05 | 0.958 | .026 | .001 |
| Internalizing | 0.017 | 1.21 | 0.227 | .047 | .017 |
| Externalizing | 0.102 | 7.27 | < .001* | .044 | .101 |
| Oppositional defiance | 0.085 | 6.02 | < .001* | .022 | .084 |
| Conduct disorder | 0.112 | 7.98 | < .001* | .023 | .111 |
| Attention problems | 0.111 | 7.88 | < .001* | .026 | .109 |
| ADHD | 0.131 | 9.32 | < .001* | .023 | .129 |
| Academic performance | -0.085 | -5.99 | < .001* | .003 | -.086 |
| Sleep quantity | -0.136 | -10.13 | < .001* | .003 | -.140 |
| Sleep quality | 0.076 | 5.37 | < .001* | .035 | .075 |
| Num. of close m. friends | 0.084 | 5.90 | < .001* | .011 | .082 |
| Num. of close f. friends | 0.063 | 4.39 | < .001* | .034 | .061 |

*Note*. Starred regressions are significant at Bonferroni corrected alpha .004. ADHD = attention/deficit hyperactivity disorder; Num. of Close M. Friends = number of close friends who are male; Num. of Close F. Friends = number of close friends who are female.

sex—demonstrate that generally, both screen time and sex predict the outcome variables, but that the effect of screen time on the outcome variables often does not depend on sex, and vice versa. For the outcome measures with non-significant interaction terms but significant main effects of both/either screen time and/or sex, it appears that screen time and sex are independent predictors of the outcome measure. For these outcome measures, the effect of either screen time or sex on the outcome variable did not depend on the other independent variable. A potential reason for that finding could be sex differences in how screens are being used. The only outcome measure demonstrating a significant interaction term, for Part 1 and for Part 2, is number of close friends who are males. It is possible that, because males in this study tend to use screen time for video gaming—which is often a social activity—more than females do (refer to Table 1), screen time and sex interact such that the effect of screen time (e.g., using screens for video gaming) on number of close male friends depends on the sex of the participant, where male participants who spend more time on screens video gaming have more male friends.

Screen time—above and beyond both SES and race/ethnicity—is a significant predictor of some internalizing symptoms, behavioral problems, academic performance, sleep quality and quantity, and the strength of peer relationships for 9- to 10-year-old children, in both boys and girls. However, the effect of screen time was small (<2% of the variance explained) for all outcomes, with SES—which was demonstrated to be a significant predictor for the nearly all outcome variables of interest—accounting for much more of the variance (~5%), perhaps because parent SES contributes to nearly every facet of children's physical and mental health outcomes [28]. Taken together, our results imply that too much time spent on screens is associated with poorer mental health, behavioral health, and academic outcomes in 9- and 10- year old children, but that negative impact on the subjects is likely not clinically harmful at this age.

The significant association between screen time and externalizing disorder symptoms was in line with previous research [13]. However, this association is not necessarily causal; for example, it has been suggested that parents/guardians of children who display externalizing disorder symptoms, along with oppositional defiance disorder and conduct disorder, are more likely to place their child in front of a screen as a distraction [29], so it is possible that externalizing disorder symptoms feed into additional screen time rather than the reverse.

The negative association between screen time and academic performance may be of some concern to parents; another group of researchers reported a similar trend in a sample of Chinese adolescents [30]. We speculate that more time dedicated to recreational screen use detracts from time spent on schoolwork and studying for exams, though this proposed explanation should be examined further. In data collection for the ABCD Study, academic screen time (e.g., using a computer to complete an academic paper) was not recorded; it is possible that academic screen time could be positively associated with academic performance, suggesting, as previous studies [22, 23] point out, that the type of screen time use is more important to consider than screen time itself.

The negative association between screen time and amount of sleep has been demonstrated previously [17] and, as in the case of academic performance, it is possible that time on screens takes away from time asleep. The positive association between sleep disorder score and screen time is of interest, though how that relationship is mediated is a topic of future research. It could be that when children and adolescents struggle with sleep, they turn to electronic media as a way to distract themselves or in an attempt to lull themselves back to sleep, or that screen use contributes to delayed bedtime, as has been suggested in previous literature [17].

The lack of significant relationships between screen time and internalizing disorder symptoms (i.e., depression and anxiety) was surprising and does not align with prior findings by researchers who also used the ABCD study to examine screen time as a predictor variable. To examine the discrepancy, we conducted a replication of their study [11], using the early release data of 4528 participants, which is less than half the sample size used in the current study. We replicated their findings closely, which suggests that the discrepancy in our results primarily arises from the differences in the sample as it doubled in size. Overall, both the current study and the previous [11] find only weak associations of screen time with internalizing problems in the baseline ABCD sample. It is possible that because internalizing disorders typically develop throughout childhood and adolescence [31, 32], 9- and 10- year old children are simply not displaying immediately noticeable internalizing symptoms.

The finding that more screen time is associated with a greater number of close friends, both male and female, is in line with previous research [21] and suggests that when on screens, adolescents are communicating with their friends via texting, social media, or video chat, and the social nature of such screen time use strengthens relationships between peers and allows them to stay connected even when apart.

The current study is not without limitations. Because participants are 9 and 10, they simply are not using screens as much as their older peers; means for screen time use are low, especially for texting and social media, two aspects of screen time that may have the most impact on peer relationships and mental health outcomes [21]. The frequencies of mature gaming and viewing of R-rated movies are also low. Similarly due to the age of the sample, the majority of participants do not display signs of mental ill health. Follow-up interview studies conducted as the sample ages would likely be more powered as adolescents increase in their screen use and they evidence more mental health issues at older ages. Beneficially, however, the longitudinal nature of the ABCD Study will allow continuation of study of these potential associations over the course of the participants' adolescence. Next, the measures used by the ABCD Study at baseline have some limitations. By restricting the screen time maximum label to "4+ hours" for all subsets of screen time apart from total screen time, it was not possible to examine extremes in screen time (e.g., the present data do not differentiate between four hours of texting and 15 hours. Additionally, the majority of outcome measures were evaluated through parent report rather than child self-report, and it is possible that parent evaluations are inaccurate, especially for more subtle symptoms such as internalizing problems. However, for the majority of outcome variables, parents responded to the Child Behavior Checklist, which demonstrates strong psychometric validity [33]. Additionally, parent report is preferred for assessing some outcome measures of interest; in externalizing problems and attention problems specifically, the positive illusory bias skews youth self-report to overly positive reports of their performance in comparison to criteria that reflects actual performance [34, 35].

## Conclusions

Both weekday and weekend total screen time are moderately associated with greater behavioral problems including ADHD, poor academic performance and poor sleep quantity and quality. Conversely, screen time is positively associated with the quantity and quality of peer relationships. The effect of screen time on those outcome measures typically does not depend on sex. Observed effect sizes are small (<2% variance explained), with SES contributing much more to the variance in outcomes. Though these associations should be monitored and examined further as this study cohort ages in mid- and late- adolescence, our results are in line with a recent review [22]. It seems that screen time itself is not strongly associated with adverse outcomes in 9- and 10- year old children.

## Supporting information

**S1 Table. Weekday and weekend differences on screen time measures.** *Note*. Significance at .05. Means and SD for weekday/weekend screen time measures given in hours. (DOCX)

**S2 Table. Correlations by sex between all variables for Part 1, weekday screen time.** *Note*. Male participant correlations are shown across the top and in upper right triangle, female participant correlations are shown along the left side and in the lower left triangle. Grayed correlations were not significant at alpha .05. Abbreviations: Tot. = total, PR = parent report, TV = television and movies, Vid. = videos, VC = video chat, Text = texting, SM = social media, VG = video games, MG = mature games, RM = R-rated movies, Dep. = depression, Anx. = anxiety, Int. = internalizing problems, Ext. = externalizing problems, ODD = oppositional defiance disorder, CD = conduct disorder, Attn. = attention problems, AD = ADHD, AP = academic performance, ST = sleep quantity, SD = sleep quality, NCB = number of close

friends who are boys, NCG = number of close friends who are girls.
(DOCX)

**S3 Table. Multicollinearity statistics (VIF and tolerance) for Part 1 variables.** *Note.* T2
Weekday = multicollinearity statistics corresponding to the interaction analyses conducted in
Table 2. T3 Males = multicollinearity statistics corresponding to the regression analyses con-
ducted in Table 3 for males only. T3 Females = multicollinearity statistics corresponding to the
regression analyses conducted in Table 3 for females only. ST = screen time. R/E = race/ethnic-
ity. SES = socioeconomic status. VIF = variance inflation factor. Tol = tolerance. Oppos.
Def = oppositional defiance disorder. Cond. Dis = conduct disorder. Attn. Prob. = attention
problems. Sleep Quant. = sleep quantity in hours. Sleep Qual. = sleep quality. Num. M. Fr. =
number of close male friends. Num. F. Fr. = number of close female friends.
(DOCX)

**S4 Table. Correlations by sex between all variables for Part 2, weekend screen time.** *Note.*
Male participant correlations are shown across the top and in upper right triangle, female par-
ticipant correlations are shown along the left side and in the lower left triangle. Grayed correla-
tions were not significant at alpha .05. Abbreviations: Tot. = total, PR = parent report,
TV = television and movies, Vid. = videos, VC = video chat, Text = texting, SM = social media,
VG = video games, MG = mature games, RM = R-rated movies, Dep. = depression, Anx. =
anxiety, Int. = internalizing problems, Ext. = externalizing problems, ODD = oppositional
defiance disorder, CD = conduct disorder, Attn. = attention problems, AD = ADHD,
AP = academic performance, ST = sleep quantity, SD = sleep quality, NCB = number of close
friends who are boys, NCG = number of close friends who are girls.
(DOCX)

**S5 Table. Multicollinearity statistics (VIF and tolerance) for Part 2 variables.** *Note.* T4
Weekend = multicollinearity statistics corresponding to the interaction analyses conducted in
Table 4. T5 Males = multicollinearity statistics corresponding to the regression analyses con-
ducted in Table 5 for males only. T5 Females = multicollinearity statistics corresponding to the
regression analyses conducted in Table 5 for females only. ST = screen time. R/E = race/ethnic-
ity. SES = socioeconomic status. VIF = variance inflation factor. Tol = tolerance. Oppos.
Def = oppositional defiance disorder. Cond. Dis = conduct disorder. Attn. Prob. = attention
problems. Sleep Quant. = sleep quantity in hours. Sleep Qual. = sleep quality. Num. M. Fr. =
number of close male friends. Num. F. Fr. = number of close female friends.
(DOCX)

**S6 Table. Depression regressed on various types of weekday screen time for Part 1, control-
ling for SES and race/ethnicity, separated by sex.** *Note.* Starred regressions are significant at
alpha .05.
(DOCX)

**S7 Table. Anxiety regressed on various types of weekday screen time for Part 1, controlling
for SES and race/ethnicity, separated by sex.** *Note.* Starred regressions are significant at alpha
.05.
(DOCX)

**S8 Table. Internalizing symptoms regressed on various types of weekday screen time for
Part 1, controlling for SES and race/ethnicity, separated by sex.** *Note.* Starred regressions
are significant at alpha .05.
(DOCX)

**S9 Table. Externalizing symptoms regressed on various types of weekday screen time for Part 1, controlling for SES and race/ethnicity, separated by sex.** *Note*. Starred regressions are significant at alpha .05.
(DOCX)

**S10 Table. Oppositional defiance disorder regressed on various types of weekday screen time for Part 1, controlling for SES and race/ethnicity, separated by sex.** *Note*. Starred regressions are significant at alpha .05.
(DOCX)

**S11 Table. Conduct disorder regressed on various types of weekday screen time for Part 1, controlling for SES and race/ethnicity, separated by sex.** *Note*. Starred regressions are significant at alpha .05.
(DOCX)

**S12 Table. Attention problems regressed on various types of weekday screen time for Part 1, controlling for SES and race/ethnicity, separated by sex.** *Note*. Starred regressions are significant at alpha .05.
(DOCX)

**S13 Table. ADHD regressed on various types of weekday screen time for Part 1, controlling for SES and race/ethnicity, separated by sex.** *Note*. Starred regressions are significant at alpha .05.
(DOCX)

**S14 Table. Grades regressed on various types of weekday screen time for Part 1, controlling for SES and race/ethnicity, separated by sex.** *Note*. Starred regressions are significant at alpha .05.
(DOCX)

**S15 Table. Average nightly hours of sleep regressed on various types of weekday screen time for Part 1, controlling for SES and race/ethnicity, separated by sex.** *Note*. Starred regressions are significant at alpha .05.
(DOCX)

**S16 Table. Sleep disorder score regressed on various types of weekday screen time for Part 1, controlling for SES and race/ethnicity, separated by sex.** *Note*. Starred regressions are significant at alpha .05.
(DOCX)

**S17 Table. Number of close friends who are boys regressed on various types of weekday screen time for Part 1, controlling for SES and race/ethnicity, separated by sex.** *Note*. Starred regressions are significant at alpha .05.
(DOCX)

**S18 Table. Number of close friends who are girls regressed on various types of weekday screen time for Part 1, controlling for SES and race/ethnicity, separated by sex.** *Note*. Starred regressions are significant at alpha .05.
(DOCX)

**S19 Table. Depression regressed on various types of weekend screen time for Part 2, controlling for SES and race/ethnicity, separated by sex.** *Note*. Starred regressions are significant at alpha .05.
(DOCX)

**S20 Table. Anxiety regressed on various types of weekend screen time for Part 2, controlling for SES and race/ethnicity, separated by sex.** *Note*. Starred regressions are significant at alpha .05.
(DOCX)

**S21 Table. Internalizing symptoms regressed on various types of weekend screen time for Part 2, controlling for SES and race/ethnicity, separated by sex.** *Note*. Starred regressions are significant at alpha .05.
(DOCX)

**S22 Table. Externalizing symptoms regressed on various types of weekend screen time for Part 2, controlling for SES and race/ethnicity, separated by sex.** *Note*. Starred regressions are significant at alpha .05.
(DOCX)

**S23 Table. Oppositional defiance disorder regressed on various types of weekend screen time for Part 2, controlling for SES and race/ethnicity, separated by sex.** *Note*. Starred regressions are significant at alpha .05.
(DOCX)

**S24 Table. Conduct disorder regressed on various types of weekend screen time for Part 2, controlling for SES and race/ethnicity, separated by sex.** *Note*. Starred regressions are significant at alpha .05.
(DOCX)

**S25 Table. Attention problems regressed on various types of weekend screen time for Part 2, controlling for SES and race/ethnicity, separated by sex.** *Note*. Starred regressions are significant at alpha .05.
(DOCX)

**S26 Table. ADHD regressed on various types of weekend screen time for Part 2, controlling for SES and race/ethnicity, separated by sex.** *Note*. Starred regressions are significant at alpha .05.
(DOCX)

**S27 Table. Grades regressed on various types of weekend screen time for Part 2, controlling for SES and race/ethnicity, separated by sex.** *Note*. Starred regressions are significant at alpha .05.
(DOCX)

**S28 Table. Average nightly hours of sleep regressed on various types of weekend screen time for Part 2, controlling for SES and race/ethnicity, separated by sex.** *Note*. Starred regressions are significant at alpha .05.
(DOCX)

**S29 Table. Sleep disorder score regressed on various types of weekend screen time for Part 2, controlling for SES and race/ethnicity, separated by sex.** *Note*. Starred regressions are significant at alpha .05.
(DOCX)

**S30 Table. Number of close friends who are boys regressed on various types of weekend screen time for Part 2, controlling for SES and race/ethnicity, separated by sex.** *Note*.

Starred regressions are significant at alpha .05.
(DOCX)

**S31 Table. Number of close friends who are girls regressed on various types of weekend screen time for Part 2, controlling for SES and race/ethnicity, separated by sex.** *Note.* Starred regressions are significant at alpha .05.
(DOCX)

## Author Contributions

**Conceptualization:** Katie N. Paulich, J. Megan Ross, Jeffrey M. Lessem, John K. Hewitt.

**Data curation:** Katie N. Paulich.

**Formal analysis:** Katie N. Paulich.

**Software:** J. Megan Ross, Jeffrey M. Lessem.

**Supervision:** John K. Hewitt.

**Validation:** Katie N. Paulich.

**Visualization:** Katie N. Paulich.

**Writing – original draft:** Katie N. Paulich.

**Writing – review & editing:** Katie N. Paulich, J. Megan Ross, Jeffrey M. Lessem, John K. Hewitt.

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
