## [Decision Letter · Decision Letter 0]

22 Mar 2021

PONE-D-20-40267

Screen time is only modestly associated with mental health, academic outcomes, and peer relationships in the Adolescent Brain Cognitive Development ℠ Study

PLOS ONE

Dear Dr. Paulich,

Thank you for submitting your manuscript to PLOS ONE. After careful consideration, we feel that it has merit but does not fully meet PLOS ONE’s publication criteria as it currently stands. Therefore, we invite you to submit a revised version of the manuscript that addresses the points raised during the review process. 

There are a number of concerns raised by the reviewers those need to be addressed before taking final decision. 

We look forward to receiving your revised manuscript.

Kind regards,

Enamul Kabir

Academic Editor

PLOS ONE

Journal Requirements:

2a) If there are ethical or legal restrictions on sharing a de-identified data set, please explain them in detail (e.g., data contain potentially sensitive information, data are owned by a third-party organization, etc.) and who has imposed them (e.g., an ethics committee). Please also provide contact information for a data access committee, ethics committee, or other institutional body to which data requests may be sent.

2b) If there are no restrictions, please upload the minimal anonymized data set necessary to replicate your study findings as either Supporting Information files or to a stable, public repository and provide us with the relevant URLs, DOIs, or accession numbers. For a list of acceptable repositories, please see http://journals.plos.org/plosone/s/data-availability#loc-recommended-repositories.

Reviewers' comments:

Reviewer's Responses to Questions

**Comments to the Author**

1. Is the manuscript technically sound, and do the data support the conclusions?

Reviewer #1: Yes

Reviewer #2: Yes

Reviewer #3: Yes

2. Has the statistical analysis been performed appropriately and rigorously? 

Reviewer #1: No

Reviewer #2: I Don't Know

Reviewer #3: No

3. Have the authors made all data underlying the findings in their manuscript fully available?

Reviewer #1: No

Reviewer #2: Yes

Reviewer #3: Yes

4. Is the manuscript presented in an intelligible fashion and written in standard English?

Reviewer #1: Yes

Reviewer #2: Yes

Reviewer #3: Yes

5. Review Comments to the Author

Reviewer #1: Comments to Manuscript Number PONE-D-20-40267

Full Title: Screen time is only modestly associated with mental health, academic outcomes, and peer relationships in the Adolescent Brain Cognitive Development ℠ Study

Short Title: Screen time and the ABCD Study

The paper examines an interesting and useful topic related to screen time and mental health, behavioral problems, academic performance, sleep habits, and peer relationships in the USA. It does fit the scope of PLoS ONE. Overall, it contributes to the advancement of knowledge and debate on matters of mental anxiety, depression, and academic results by those who are using more screens in the United States. Though the study finding got a small effect size. But still, the results are significant in line with the expected hypothesis. The findings will help to understand to what extent screen time are creating vulnerability to the child during their early adolescent stage.

However, the fundamental shortcomings of the paper are:

1. Why the study is divided by study 1 and study 2 in the results section. I understand study 1 is for weekday screen time and study 2 for weekend screen time. Changing it to part 1 and part 2 would help. Need a clear justification for dividing these two parts.

2. For logistics regression, the study divides the sample by their sex. The study runs two separate regressions for both males and females. Some further explanation and justification need to be provided regarding the two separate regressions. It would be interesting if the author(s) could add a combined regression (for both male and female) using sex as a dummy explanatory variable and then run two separate regression for male and female to check how does gender play a role in explaining the effect of screen time on mental health, academic performance and peer relationships. The study is restricting the sample for two separate regression by their sex and in fact, sex is a channel to explain the role of screen time on different outcome variables.

3. The study found a very low effect size to explain the linkage between total screen time and the outcome measures. Along with the total screen time, the study can estimate the effect of screen time by creating a dummy for an acceptable level of screen time and beyond that. This will help to show how too much screen time creating an effect on the outcome variables.

4. The descriptive statistics about the outcome and explanatory variables should be provided in the main paper instead of placing them in the appendix. Suggest bringing table S1 and S2 after combining them in a single table up into the main body of the text so the reader has a better sense of their characteristics (if the appendix is published with the paper (and not just online) that may be less of a problem).

5. It would be better to provide the overall descriptive statistics about the SES.

6. Reference to other studies, including Twenge and Campbell, 2018; Oswald et al., 2020.

7. Better to use the clustering by different demographic zone while running a regression. As the study participant covers different demographic zone, therefore, the clustering (clustered standard errors) could provide more robust results.

8. The discussion section is written appropriately. However, the results section is not written consistently. It would be better to make the writing of the result section consistent to make it more reader-friendly.

9. Though the study focused too much on sex and weekday/weekend without proper justification. A clear justification is useful to add.

10. The study considers many outcomes without focusing on them in more detail. It would be better if the study restricts their outcome variable and then cover the heterogeneous channels to find the linkage between the outcome and screen time. For example, sex is a channel where the effect size of screen time on the outcome variables is different depending on the sex of the participants. The study could concentrate on some other channels from the SES to find the effect of screen time on the outcome variables in more detail.

Reviewer #2: I enjoyed reading this article and drawing conclusion from larger sample size is commendable. Also the study pointed out the influence of screen time on academic outcomes and others which is very insightful. However, I have few comments and suggestions for them.

The title “Screen time is only modestly associated with mental health, academic outcomes, and peer relationships in the Adolescent Brain Cognitive Development” should be reshaped. A good title should at least, tell us the dependent and independent variables, study population and the area of study. The title looks a bit confusing.

Abstract section

“We are using screens more than ever” [line 23]. This is not clear. Please the “we” should be clarified. Who are you referring to?

Main Text

Introduction

“with 95% of teens having access to a smartphone”…. [line 45]. The “95%”, is it a global prevalence or what?

The authors did well by stating the expected results/working hypothesis. However, the study lacks theoretical conceptual framework. Therefore, I suggest the authors should add a theory to it.

Also, authors should tell us the prevalence of screen time for us to be clear about proportion of children being exposed to screen, from global to study area perspective, if such data exist. Such trend analysis could enrich the paper.

Statistical analysis

Why should the authors use Multiple linear regressions because such estimating technique may not help to understand differences within groupings? Also, they fail to account or check for multicollinearity that might exist between explanatory variables.

Also, they should simply tell us how the results were interpreted.

Discussion section

The authors did a great job by comparing their results with previous studies. However, their explanations were mostly based on conjecture and speculations without literature. I suggest the authors adopt/adapt a theory and situate their results and discussions in the theory.

Also, at the introductory aspect of the discussion, I suggest the authors should tell us the main/key findings in brief and show us how significant are these results before moving on to discuss them.

References

The authors also used current literature which is commendable.

Overall, the paper could be published if they are able to improve the paper.

Also, they should proof read for few grammatical errors.

Reviewer #3: 1) Title needs to be shortened too long.

2) Introduction and literature should be given under separate titles.

3) The importance of this study should be explained in more detail. Research questions should be specified more clearly.

4) The current study is divided into Study 1 and Study 2. The reason for this was explained as “We divided the current study into two studies to better assess fundamental differences in anticipated weekday and weekend screen time use. On an average weekday, children aged 9 and 10 are likely to be in a structured educational environment and, therefore, limited in their daytime screen use. On weekends, children are likely to be at home or in differently structured environments and may have ready access to screens.”. This explanation is not very satisfactory. A scientific explanation is required. What kind of trouble would it have caused if seven days were taken together and evaluated?

5) “Bivariate Pearson correlations”, “Independent samples t test” and “Multiple linear regressions” analyzes were applied. One of the basic assumptions of these analyzes is the normal distribution assumption. No information was given in the article that the normal distribution assumption was met.

6) There is no need to give confidence intervals in the tables. Constant and non-standardized beta coefficients should also be given in the tables of regression models.

6. PLOS authors have the option to publish the peer review history of their article (what does this mean?). If published, this will include your full peer review and any attached files.

Reviewer #1: No

Reviewer #2: No

Reviewer #3: No

---

## [Author Response · Author response to Decision Letter 0]

5 May 2021

Response to Reviewers

We appreciate the opportunity to revise and resubmit our manuscript; incorporating the reviewer suggestions below has improved the quality of our paper, and we thank you for the additional consideration. 

*Please note that all references to line number are within the Revised Manuscript with Track Changes.

Journal Requirements:

To the best of our knowledge, our manuscript meets PLOS ONE’s style requirements. 

2. We note that you have indicated that data from this study are available upon request. PLOS only allows data to be available upon request if there are legal or ethical restrictions on sharing data publicly.

2a) If there are ethical or legal restrictions on sharing a de-identified data set, please explain them in detail (e.g., data contain potentially sensitive information, data are owned by a third-party organization, etc.) and who has imposed them (e.g., an ethics committee). Please also provide contact information for a data access committee, ethics committee, or other institutional body to which data requests may be sent.

The following has been added to the revised cover letter: “Legal restrictions on sharing the utilized de-identified data set are in place; the data are owned by a third-party organization, the National Institute of Mental Health Data Archive. However, qualified researchers can request access to ABCD shared data at https://nda.nih.gov/abcd/request-access. The datasets used in this study can be found in online repositories; the ABCD data used in this paper came from NIMH Data Archive Digital Object Identifier 10.15154/1519271.”

Additionally, the following has been added to the revised manuscript at line 173: “The data used in this study are owned by the National Institute of Mental Health Data Archive; qualified researchers can request access to ABCD shared data at https://nda.nih.gov/abcd/request-access. The ABCD data used in this paper came from NIMH Data Archive Digital Object Identifier 10.15154/1519271.”

Reviewer #1: Comments to Manuscript Number PONE-D-20-40267

The paper examines an interesting and useful topic related to screen time and mental health, behavioral problems, academic performance, sleep habits, and peer relationships in the USA. It does fit the scope of PLoS ONE. Overall, it contributes to the advancement of knowledge and debate on matters of mental anxiety, depression, and academic results by those who are using more screens in the United States. Though the study finding got a small effect size. But still, the results are significant in line with the expected hypothesis. The findings will help to understand to what extent screen time are creating vulnerability to the child during their early adolescent stage.

However, the fundamental shortcomings of the paper are:

1. Why the study is divided by study 1 and study 2 in the results section. I understand study 1 is for weekday screen time and study 2 for weekend screen time. Changing it to part 1 and part 2 would help. Need a clear justification for dividing these two parts.

The following has been added to the end of the Participants section at line 185: “We divided the current study into two parts to better assess fundamental differences in weekday and weekend screen time use. There exists a significant difference in time spent on screens, t(11,832) = -52.31, p<.001, during a weekday (M = 3.46, SD = 3.10) and time spent on screens during the weekend (M = 4.62, SD = 3.63). There was also a significant difference in parent reports of their child’s screen time, t(11,747) = -61.63, p<.001, between weekdays (M = 2.55, SD = 2.59) and weekends (M = 3.99, SD = 2.66). Additionally, there existed significant differences between weekday and weekend screen usage type (e.g., TV/movies, online videos, gaming, etc.) for every usage type examined (see Table S1). These significant differences suggest that weekday and weekend screen time and screen time use type differ, and therefore, should be examined separately.”

We have followed the suggestion to change “Study 1” and “Study 2” to “Part 1” and “Part 2” to make the division more clear.

2. For logistics regression, the study divides the sample by their sex. The study runs two separate regressions for both males and females. Some further explanation and justification need to be provided regarding the two separate regressions. It would be interesting if the author(s) could add a combined regression (for both male and female) using sex as a dummy explanatory variable and then run two separate regression for male and female to check how does gender play a role in explaining the effect of screen time on mental health, academic performance and peer relationships. The study is restricting the sample for two separate regression by their sex and in fact, sex is a channel to explain the role of screen time on different outcome variables.

Regarding justification for two separate regressions, the following has been added to the end of the Statistical Analysis section of the Materials and Method at line 345: “We conducted analyses separately by sex because there existed significant sex differences in total weekday screen time, t(11,831) = 10.22, p<.001, with males (M = 3.74, SD = 3.17) spending more weekday time on screens than females (M = 3.16, SD = 2.99); in total weekend screen time, t(11,829) = 13.54, p<.001, with males (M = 5.05, SD = 3.68) spending more weekend time on screens than females (M = 4.16, SD = 3.53); as well as each outcome measure. The sex differences suggested that males and females differed in both independent and dependent variables, and therefore, should be examined separately. Subsequent analyses were conducted separately by sex. Table 1 provides sex differences separated by Parts 1 and 2.”

Additionally, we followed the suggestion to add a combined regression, using sex as a dummy variable and as an interaction term to investigate whether the effect of screen time depends on sex. The following was inserted into the Statistical Analysis section of the Materials and Methods, at line 333: “We conducted a combined regression (across sex), coding sex as a dummy variable to investigate—via interaction test—whether the effect of screen time on our outcome variables depended on sex.”

Because the study is divided into Part 1 and Part 2, we conducted two additional regressions in this manner (one for Part 1 and one for Part 2), as weekday and weekend screen time use differs. The results of these additional analyses can be found in the new Table 2 and Table 4.

3. The study found a very low effect size to explain the linkage between total screen time and the outcome measures. Along with the total screen time, the study can estimate the effect of screen time by creating a dummy for an acceptable level of screen time and beyond that. This will help to show how too much screen time creating an effect on the outcome variables.

Thank you for this suggestion. Unfortunately, there is not currently an empirically established threshold for an “acceptable level of screen time and beyond,” so to create such a dummy code in the context of this study would be an arbitrary decision based only on speculation and not on previous literature. We understand the reasoning behind this suggestion, and agree that it would be a promising avenue for future research on this topic, once such a threshold has been firmly established in the literature on screen time.

4. The descriptive statistics about the outcome and explanatory variables should be provided in the main paper instead of placing them in the appendix. Suggest bringing table S1 and S2 after combining them in a single table up into the main body of the text so the reader has a better sense of their characteristics (if the appendix is published with the paper (and not just online) that may be less of a problem).

We have added Table 1, which displays descriptive statistics and sex differences, into the main body of the text to address this concern. As the reviewer suggested, this table is a combined Table S1 and S2.

5. It would be better to provide the overall descriptive statistics about the SES.

Overall descriptive statistics about SES have been added to the end of the Participants section of the Materials and Methods, at line 174, and have been removed from the Part 1 and Part 2 sections for the sake of brevity. 

6. Reference to other studies, including Twenge and Campbell, 2018; Oswald et al., 2020.

References to both Twenge & Campbell, 2018, and Oswald et al., 2020, have been added to the Introduction, and both studies have been cited in the references. Thank you for bringing these relevant studies to our attention.

7. Better to use the clustering by different demographic zone while running a regression. As the study participant covers different demographic zone, therefore, the clustering (clustered standard errors) could provide more robust results.

The regressions are conducted by sex, clustering the results by that demographic zone. We consider this study to be an initial exploration of the relationships between screen time and those various aspects of well-being and health in 9- and 10- year old children, a population that is seldom examined in studies of screen time on adolescence. As such, we have controlled for SES in every regression, rather than dividing it into different zones as suggested, in order to glean an overview of the findings. Standard errors have been added to the tables, in place of confidence intervals.

8. The discussion section is written appropriately. However, the results section is not written consistently. It would be better to make the writing of the result section consistent to make it more reader-friendly.

Thank you for your comment; the writing of the Results section has been revised accordingly.

9. Though the study focused too much on sex and weekday/weekend without proper justification. A clear justification is useful to add.

The following has been added to the end of the Participants section at line 185: “We divided the current study into two parts to better assess fundamental differences in weekday and weekend screen time use. There exists a significant difference in time spent on screens, t(11,832) = -52.31, p<.001, during a weekday (M = 3.46, SD = 3.10) and time spent on screens during the weekend (M = 4.62, SD = 3.63). There was also a significant difference in parent reports of their child’s screen time, t(11,747) = -61.63, p<.001, between weekdays (M = 2.55, SD = 2.59) and weekends (M = 3.99, SD = 2.66). Additionally, there existed significant differences between weekday and weekend screen usage type (e.g., TV/movies, online videos, gaming, etc.) for every usage type examined (see Table S1). These significant differences suggest that weekday and weekend screen time and screen time use type differ, and therefore, should be examined separately.”

The following has been added to the end of the Statistical Analysis section of the Materials and Method at line 345: “We conducted analyses separately by sex because there existed significant sex differences in total weekday screen time, t(11,831) = 10.22, p<.001, with males (M = 3.74, SD = 3.17) spending more weekday time on screens than females (M = 3.16, SD = 2.99); in total weekend screen time, t(11,829) = 13.54, p<.001, with males (M = 5.05, SD = 3.68) spending more weekend time on screens than females (M = 4.16, SD = 3.53); as well as each outcome measure. The sex differences suggested that males and females differed in both independent and dependent variables, and therefore, should be examined separately. Subsequent analyses were conducted separately by sex. Table 1 provides sex differences separated by Parts 1 and 2.”

10. The study considers many outcomes without focusing on them in more detail. It would be better if the study restricts their outcome variable and then cover the heterogeneous channels to find the linkage between the outcome and screen time. For example, sex is a channel where the effect size of screen time on the outcome variables is different depending on the sex of the participants. The study could concentrate on some other channels from the SES to find the effect of screen time on the outcome variables in more detail.

Thank you for this suggestion; our study considers many outcomes without focusing on specific ones in more detail because we consider this study to be an initial exploration of the relationships between screen time and those various aspects of well-being and health in 9- and 10- year old children, a population that is seldom examined in studies of screen time on adolescence. Follow-up studies should focus on more specific variables of interest and additional channels such as SES, which we have controlled for in this study to examine effects of screen time above and beyond SES. 

We did follow suggestion #2, adding an interacting effect of sex, to explore that specific relationship further.

Reviewer #2: I enjoyed reading this article and drawing conclusion from larger sample size is commendable. Also the study pointed out the influence of screen time on academic outcomes and others which is very insightful. However, I have few comments and suggestions for them.

The title “Screen time is only modestly associated with mental health, academic outcomes, and peer relationships in the Adolescent Brain Cognitive Development” should be reshaped. A good title should at least, tell us the dependent and independent variables, study population and the area of study. The title looks a bit confusing.

The title of the manuscript has been revised to: Screen time and early adolescent mental health, academic, and social outcomes in 9- and 10- year old children: Utilizing the Adolescent Brain Cognitive Development ℠ (ABCD) Study

The title lists the dependent variables: “early adolescent mental health, academic, and social outcomes” 

The title names the independent variable: “screen time”

The title names the study population: “the Adolescent Brain Cognitive Development ℠ (ABCD) Study” and “9- and 10- year old children”

The title names the area of study: “early adolescent”, “screen time,” and “9- and 10- year old children”

Abstract section

“We are using screens more than ever” [line 23]. This is not clear. Please the “we” should be clarified. Who are you referring to?

That line (now line 30) has been revised to read, “In a technology-driven society, screens are being used more than ever.”

Main Text

Introduction

“with 95% of teens having access to a smartphone”…. [line 45]. The “95%”, is it a global prevalence or what?

That line (now line 51) has been revised to read: “Children and adolescents are spending more time on screens and electronic media than ever before, with 95% of teens in the United States having access to a smartphone.”

The authors did well by stating the expected results/working hypothesis. However, the study lacks theoretical conceptual framework. Therefore, I suggest the authors should add a theory to it.

The final paragraph of the Introduction at line 135 now reads, “Given the previous findings on screen time associations, we ask: in 9- and 10- year old children, what relationships exist between screen time and mental health, behavioral health, academic success, and peer relationships? We hypothesized that total screen time would be 1) positively associated with increased depression and anxiety symptoms as well as behavioral problems including ADHD, 2) negatively associated with academic performance and sleep quantity and quality, and 3) positively associated with quantity and quality of peer relationships. Our study is unique in its ability to allow us to determine the magnitude of these associations, their importance, and potential adverse impacts of increased screen time in a novel and very large, diverse national sample of 9- to 10- year old children. Our findings lay groundwork for future analyses on the longitudinal ABCD Study sample.”

Also, authors should tell us the prevalence of screen time for us to be clear about proportion of children being exposed to screen, from global to study area perspective, if such data exist. Such trend analysis could enrich the paper.

The beginning of the Introduction now reads: “Children and adolescents are spending more time on screens and electronic media than ever before, with 95% of teens in the United States having access to a smartphone [1]. While global inequalities in technology use certainly exist—in 71 out of 195 countries globally, less than half the population has access to the internet—it is undeniable that average global technology use is on the rise, especially among youth [2].”

Statistical analysis

Why should the authors use Multiple linear regressions because such estimating technique may not help to understand differences within groupings? Also, they fail to account or check for multicollinearity that might exist between explanatory variables.

We utilized Multiple Regression in order to investigate general predictive power of screen time on various adolescent well-being outcomes (e.g., depression, anxiety, conduct disorder, strength of peer relationships, etc.). We grouped the regressions by sex, and also examined sex as an interaction with screen time, to investigate the demonstrated sex differences more clearly. Our data met the assumptions of a Multiple Regression analysis. The data did not demonstrate collinearity or multicollinearity, as shown by the correlation tables S2 and S3, which display correlations between all variables.

To clarify the question of multicollinearity, the following was added to the Results section for each Part (following referral to Tables S2 and S3), at lines 367 and 423. “The data do not demonstrate multicollinearity.”

Also, they should simply tell us how the results were interpreted.

The first paragraph of the discussion, at line 485, has been revised and now reads: “These results have important implications. The lack of consistently significant interactions between sex and screen time—but often significant main effects for both sex and screen time—demonstrate that independently, both screen time and sex predict the outcome variables, but that the effect of screen time on the outcome variables does not depend on sex, and vice versa. Screen time—above and beyond both SES and race/ethnicity—is a significant predictor of internalizing symptoms, behavioral problems, academic performance, sleep quality and quantity, and the strength of peer relationships for 9- to 10-year-old children, in both boys and girls. However, the effect of screen time was small (<2% of the variance explained) for all outcomes, with SES accounting for much more of the variance (~5%). Taken together, our results imply that too much time spent on screens is associated with poorer mental health, behavioral health, and academic outcomes in 9- and 10- year old children, but that negative impact on the subjects is likely not clinically harmful at this age.”

Discussion section

The authors did a great job by comparing their results with previous studies. However, their explanations were mostly based on conjecture and speculations without literature. I suggest the authors adopt/adapt a theory and situate their results and discussions in the theory.

Potential explanations for our findings are supported by the following literature, which are cited in the Discussion:

17. Hale L, Guan S. Screen time and sleep among school-aged children and adolescents: a systematic literature review. Sleep Med Rev. 2015 Jun;21:50-8. doi: 10.1016/j.smrv.2014.07.007. Epub 2014 Aug 12. PMID: 25193149; PMCID: PMC4437561.

21. Iannotti RJ, Kogan MD, Janssen I, Boyce WF. Patterns of adolescent physical activity, screen-based media use, and positive and negative health indicators in the U.S. and Canada. J Adolesc Health. 2009 May; 44(5): 493–499. doi:10.1016/j.jadohealth.2008.10.142.

22. Odgers CL, Jensen MR. Annual research review: Adolescent mental health in the digital age: facts, fears, and future directions. J of Child Psychology and Psychiatry. 2020 Jan 17:61(3). 

23. Sanders T, Parker PD, Pozo-Cruz B, Noetel M, Lonsdale C. Type of screen time moderates effects on outcomes in 4013 children: evidence from the Longitudinal Study of Australian Children. Int J Behav Nutr Phys Act. 2019:16:117. doi:10.1186/s12966-019-0881-7.

28. Smith BJ, Grunseit A, Hardy LL, King L, Wolfenden L, Milat A. Parental influences on child physical activity and screen viewing time: a population based study. BMC Public Health. 2010:10(593):1-11. doi:10.1186/1471-2458-10-593.

30. Durbeej N, Sorman K, Selinus EN, Lundstrom S, Lichtenstein P, Hellner C, Halldner L. Trends in childhood and adolescent internalizing symptoms: Results from Swedish population based twin cohorts. BMC Psychology. 2019; 7:9.

31. Shanahan L, Calkins SD, Keane SP, Kelleher R, Suffness R. Trajectories of internalizing symptoms across childhood: The roles of biological self-regulation and maternal psychopathy. Dev Psychopathol. 2014 Nov; 26(4 0 2): 1353-1368.

We also found additional support for our potential explanations and cited them. Where current literature did not exist to provide potential explanations for the findings, our own possible explanations were presented as avenues for future research.

Also, at the introductory aspect of the discussion, I suggest the authors should tell us the main/key findings in brief and show us how significant are these results before moving on to discuss them.

The first paragraph of the discussion, at line 485, has been revised and now reads: “These results have important implications. The lack of consistently significant interactions between sex and screen time—but often significant main effects for both sex and screen time—demonstrate that independently, both screen time and sex predict the outcome variables, but that the effect of screen time on the outcome variables does not depend on sex, and vice versa. Screen time—above and beyond both SES and race/ethnicity—is a significant predictor of internalizing symptoms, behavioral problems, academic performance, sleep quality and quantity, and the strength of peer relationships for 9- to 10-year-old children, in both boys and girls. However, the effect of screen time was small (<2% of the variance explained) for all outcomes, with SES accounting for much more of the variance (~5%). Taken together, our results imply that too much time spent on screens is associated with poorer mental health, behavioral health, and academic outcomes in 9- and 10- year old children, but that negative impact on the subjects is likely not clinically harmful at this age.”

References

The authors also used current literature which is commendable.

Overall, the paper could be published if they are able to improve the paper.

Also, they should proof read for few grammatical errors.

The paper has been revised and improved, and has been proofread for grammatical concerns.

Reviewer #3: 

1) Title needs to be shortened too long.

The title has been revised to: Screen time and early adolescent mental health, academic, and social outcomes in 9- and 10- year old children: Utilizing the Adolescent Brain Cognitive Development ℠ (ABCD) Study

The title length is under the maximum 250 characters specified by the PLOS ONE Submission Guidelines. https://journals.plos.org/plosone/s/submission-guidelines#loc-title

2) Introduction and literature should be given under separate titles.

According to the PLOS ONE style guide (provided by PLOS ONE editors, https://journals.plos.org/plosone/s/file?id=wjVg/PLOSOne_formatting_sample_main_body.pdf) there is not a separate heading for literature.

3) The importance of this study should be explained in more detail. Research questions should be specified more clearly.

The final sentences of the Introduction section, at line 135, have been revised to read: “Given the previous findings on screen time associations, we ask: in 9- and 10- year old children, what relationships exist between screen time and mental health, behavioral health, academic success, and peer relationships? We hypothesized that total screen time would be 1) positively associated with increased depression and anxiety symptoms as well as behavioral problems including ADHD, 2) negatively associated with academic performance and sleep quantity and quality, and 3) positively associated with quantity and quality of peer relationships. Our study is unique in its ability to allow us to determine the magnitude of these associations, their importance, and potential adverse impacts of increased screen time in a novel and very large, diverse national sample of 9- to 10- year old children. Our findings lay groundwork for future analyses on the longitudinal ABCD Study sample.”

4) The current study is divided into Study 1 and Study 2. The reason for this was explained as “We divided the current study into two studies to better assess fundamental differences in anticipated weekday and weekend screen time use. On an average weekday, children aged 9 and 10 are likely to be in a structured educational environment and, therefore, limited in their daytime screen use. On weekends, children are likely to be at home or in differently structured environments and may have ready access to screens.”. This explanation is not very satisfactory. A scientific explanation is required. What kind of trouble would it have caused if seven days were taken together and evaluated?

The following has been added to the end of the Participants section at line 185: “We divided the current study into two parts to better assess fundamental differences in weekday and weekend screen time use. There exists a significant difference in time spent on screens, t(11,832) = -52.31, p<.001, during a weekday (M = 3.46, SD = 3.10) and time spent on screens during the weekend (M = 4.62, SD = 3.63). There was also a significant difference in parent reports of their child’s screen time, t(11,747) = -61.63, p<.001, between weekdays (M = 2.55, SD = 2.59) and weekends (M = 3.99, SD = 2.66). Additionally, there existed significant differences between weekday and weekend screen usage type (e.g., TV/movies, online videos, gaming, etc.) for every usage type examined (see Table S1). These significant differences suggest that weekday and weekend screen time and screen time use type differ, and therefore, should be examined separately.”

Please note that the labels “Study 1” and “Study 2” have been changed to “Part 1” and “Part 2” to reflect the adjustment.

5) “Bivariate Pearson correlations”, “Independent samples t test” and “Multiple linear regressions” analyzes were applied. One of the basic assumptions of these analyzes is the normal distribution assumption. No information was given in the article that the normal distribution assumption was met.

The following has been added to the Statistical analysis section of the Materials and methods at line 338: “The analyses conducted rely on the normal distribution assumption; the independent variables (screen time) and dependent variables are only approximately normally distributed and thus p values are necessarily subject to some imprecision.”

6) There is no need to give confidence intervals in the tables. Constant and non-standardized beta coefficients should also be given in the tables of regression models.

Rather than reporting 95% confidence intervals, the tables have been revised to instead report standard errors, which appears to be more typical.

Multiple sources suggest that providing either standardized or unstandardized Beta coefficients is acceptable.

---

## [Decision Letter · Decision Letter 1]

26 May 2021

PONE-D-20-40267R1

Screen time and early adolescent mental health, academic, and social outcomes in 9- and 10- year old children: Utilizing the Adolescent Brain Cognitive Development ℠ (ABCD) Study

PLOS ONE

Dear Dr. Paulich,

Thank you for submitting your manuscript to PLOS ONE. After careful consideration, we feel that it has merit but does not fully meet PLOS ONE’s publication criteria as it currently stands. Therefore, we invite you to submit a revised version of the manuscript that addresses the points raised during the review process.

Some of the important variables (e.g., race/ethnicity, SES)  are missing, these variables need to be added in the analysis. Check multicollinearity as the correlation between the variables are high. Proper interpretations of the interaction terms are also necessary.  

We look forward to receiving your revised manuscript.

Kind regards,

Enamul Kabir

Academic Editor

PLOS ONE

Reviewers' comments:

Reviewer's Responses to Questions

**Comments to the Author**

1. If the authors have adequately addressed your comments raised in a previous round of review and you feel that this manuscript is now acceptable for publication, you may indicate that here to bypass the “Comments to the Author” section, enter your conflict of interest statement in the “Confidential to Editor” section, and submit your "Accept" recommendation.

Reviewer #1: (No Response)

Reviewer #3: All comments have been addressed

2. Is the manuscript technically sound, and do the data support the conclusions?

Reviewer #1: (No Response)

Reviewer #3: Yes

3. Has the statistical analysis been performed appropriately and rigorously? 

Reviewer #1: (No Response)

Reviewer #3: Yes

4. Have the authors made all data underlying the findings in their manuscript fully available?

Reviewer #1: (No Response)

Reviewer #3: Yes

5. Is the manuscript presented in an intelligible fashion and written in standard English?

Reviewer #1: (No Response)

Reviewer #3: Yes

6. Review Comments to the Author

Reviewer #1: Comments to Manuscript Number PONE-D-20-40267

Full Title: Screen time and early adolescent mental health, academic, and social outcomes in 9- and 10- year old children: Utilizing the Adolescent Brain Cognitive Development ℠

(ABCD) Study

Short Title: Screen time and the ABCD Study

Thanks to the authors for addressing most of the comments that were made during the first review. Still, I have few comments to fit the paper as a good one to publish in PLoS ONE. Therefore, the paper should address the existing shortcomings:

1. How is the variable sex dummy constructed? Which one is the base category to compare with?

2. The authors have used the interaction of sex and screen time (weekday/weekend) and mostly got insignificant results. But they somehow missed the main interpretation of interaction terms. Why the coefficient of interaction is insignificant while they are highly significant separately? In the discussion section, the authors have written few sentences on this issue. However, it requires more discussion on it as the existing write-up may create confusion.

3. The paper mostly focused on sex as an explanatory variable along with screen time. But this lacks concentration on SES and race/ethnicity in results and discussion. Adding race/ethnicity and SES by creating dummies will increase the scope and contribution of this paper. As they are included in each model but not reported. Therefore, reporting them in main results, particularly, in table 2 and table 4 (of the revised submission) similar to sex would be much appreciated.

4. One of the concerns was multicollinearity. The author says there is no multicollinearity referring to the appendix table 3 where the correlation among the variables are reported including the sets of outcome and explanatory variables without reporting SES and race/ethnicity (these are included in regression table). Better to produce tables with multicollinearity tests for models used in this paper (alternative to correlation table).

Reviewer #3: I reviewed the paper. It was a good paper. The requisite modifications have been done. It can be published as it is.

7. PLOS authors have the option to publish the peer review history of their article (what does this mean?). If published, this will include your full peer review and any attached files.

Reviewer #1: No

Reviewer #3: No

---

## [Author Response · Author response to Decision Letter 1]

6 Jul 2021

We appreciate the opportunity to revise and resubmit our manuscript; incorporating the reviewer suggestions below has improved the quality of our paper, and we thank you for the additional consideration. 

*Please note that all references to line number are within the Revised Manuscript with Track Changes.

Reviewer #1: Comments to Manuscript Number PONE-D-20-40267

Full Title: Screen time and early adolescent mental health, academic, and social outcomes in 9- and 10- year old children: Utilizing the Adolescent Brain Cognitive Development ℠

(ABCD) Study

Short Title: Screen time and the ABCD Study

Thanks to the authors for addressing most of the comments that were made during the first review. Still, I have few comments to fit the paper as a good one to publish in PLoS ONE. Therefore, the paper should address the existing shortcomings:

1. How is the variable sex dummy constructed? Which one is the base category to compare with?

The following has been added to the Statistical Analysis section of the Materials and Methods at line 306: Sex was dummy coded with females = 0 and males = 1, making “females” the base category for comparison.

2. The authors have used the interaction of sex and screen time (weekday/weekend) and mostly got insignificant results. But they somehow missed the main interpretation of interaction terms. Why the coefficient of interaction is insignificant while they are highly significant separately? In the discussion section, the authors have written few sentences on this issue. However, it requires more discussion on it as the existing write-up may create confusion.

Thank you for urging clarification on these interpretations. 

The beginning of the Discussion section at line 461 now reads: These results have important implications. The lack of consistently significant interactions between screen time and sex—but often significant main effects for both screen time and sex—demonstrate that generally, both screen time and sex predict the outcome variables, but that the effect of screen time on the outcome variables often does not depend on sex, and vice versa. For the outcome measures with non-significant interaction terms but significant main effects of both/either screen time and/or sex, it appears that screen time and sex are independent predictors of the outcome measure. For these outcome measures, the effect of either screen time or sex on the outcome variable did not depend on the other independent variable. A potential reason for that finding could be sex differences in how screens are being used. The only outcome measure demonstrating a significant interaction term, for Part 1 and for Part 2, is number of close friends who are males. It is possible that, because males in this study tend to use screen time for video gaming—which is often a social activity—more than females do (refer to Table 1), screen time and sex interact such that the effect of screen time (e.g., using screens for video gaming) on number of close male friends depends on the sex of the participant, where male participants who spend more time on screens video gaming have more male friends.

3. The paper mostly focused on sex as an explanatory variable along with screen time. But this lacks concentration on SES and race/ethnicity in results and discussion. Adding race/ethnicity and SES by creating dummies will increase the scope and contribution of this paper. As they are included in each model but not reported. Therefore, reporting them in main results, particularly, in table 2 and table 4 (of the revised submission) similar to sex would be much appreciated.

Thank you for this suggestion. Both race/ethnicity and SES are already dummy coded, as they are categorical variables rather than continuous variables. 

For SES, the coding scheme is as follows: 1 = < $5,000; 2 = $5,000 - $11,999; 3 = $12,000 - $15,999; 4 = $16,000 - $24,999; 5 = $25,000 - $34,999; 6 = $35,000 - $49,999; 7 = $50,000 - $74,999; 8 = $75,000 - $99,999; 9 = $100,000 - $199,999; and 10 = $200,000+; with options for “don’t know” and “refuse to answer.” This information has been added to line 291 of the Combined family income explanation of the Measures section of the Materials and Methods.

For race/ethnicity, the coding scheme is as follows: 1 = White; 2 = Black; 3 = Hispanic; 4 = Asian; 5 = Other. This information has been added to line 297 of the Race/Ethnicity explanation of the Measures section of the Materials and Methods.

We have followed the suggestion to report statistics for both race/ethnicity and SES, as we agree that including them will increase the scope and contribution of this paper. Please refer to Table 2 and Table 4 (of the revised/latest submission) to find the reported results. 

The following has also been added to lines 349 and 408 just prior to Table 2 and Table 4: Our primary interest was examination of the effects of screen time and sex on our dependent variables; however, we also report results for race/ethnicity and SES for the sake of completeness. The main effect of SES was also often significant.

We discuss the significance of the main effect of SES in the Discussion section at line 480: However, the effect of screen time was small (<2% of the variance explained) for all outcomes, with SES—which was demonstrated to be a significant predictor for the nearly all outcome variables of interest— accounting for much more of the variance (~5%), perhaps because parent SES contributes to nearly every facet of children’s physical and mental health outcomes [28].

4. One of the concerns was multicollinearity. The author says there is no multicollinearity referring to the appendix table 3 where the correlation among the variables are reported including the sets of outcome and explanatory variables without reporting SES and race/ethnicity (these are included in regression table). Better to produce tables with multicollinearity tests for models used in this paper (alternative to correlation table).

Thank you for this suggestion. We have conducted multicollinearity tests via examination of tolerance and variance inflation factor (VIF), and have produced two new tables displaying those results for the Supporting Information, S3 Table and S5 Table. We chose to retain the correlation tables (S2 Table and S4 Table) in order to provide readers with both the multicollinearity test results as well as the correlation matrices, in case both tables are of interest to readers. The results of both the multicollinearity tests and the correlation tables show that our variables do not demonstrate multicollinearity. 

The following has also been added to the Part 1 section of the Results at line 345: The data do not demonstrate multicollinearity, as seen in S3 Table. 

The following has been added to the Part 2 section of the Results at line 403: The data do not demonstrate multicollinearity, as seen in S5 Table.

Please note that the original S3 table (showing correlations between variables for Part 2) has been renamed to S4 Table, and subsequent Supporting Information Tables have been renamed accordingly and referred to accordingly in the manuscript. The changes allow the Supporting Information Tables to be referred to in numeric order.

---

## [Decision Letter · Decision Letter 2]

11 Aug 2021

Screen time and early adolescent mental health, academic, and social outcomes in 9- and 10- year old children: Utilizing the Adolescent Brain Cognitive Development ℠ (ABCD) Study

PONE-D-20-40267R2

Dear Dr. Paulich,

We’re pleased to inform you that your manuscript has been judged scientifically suitable for publication and will be formally accepted for publication once it meets all outstanding technical requirements.

Kind regards,

Enamul Kabir

Academic Editor

PLOS ONE

Additional Editor Comments (optional):

Reviewers' comments:

Reviewer's Responses to Questions

**Comments to the Author**

1. If the authors have adequately addressed your comments raised in a previous round of review and you feel that this manuscript is now acceptable for publication, you may indicate that here to bypass the “Comments to the Author” section, enter your conflict of interest statement in the “Confidential to Editor” section, and submit your "Accept" recommendation.

Reviewer #1: All comments have been addressed

Reviewer #3: All comments have been addressed

2. Is the manuscript technically sound, and do the data support the conclusions?

Reviewer #1: Yes

Reviewer #3: Yes

3. Has the statistical analysis been performed appropriately and rigorously? 

Reviewer #1: Yes

Reviewer #3: Yes

4. Have the authors made all data underlying the findings in their manuscript fully available?

Reviewer #1: No

Reviewer #3: Yes

5. Is the manuscript presented in an intelligible fashion and written in standard English?

Reviewer #1: Yes

Reviewer #3: Yes

6. Review Comments to the Author

Reviewer #1: Thanks to the authors for their effort to address all the comments and implement the recommendations that were made during the second time review. I appreciate their work to make the manuscript reads better. I think this version now fits as a good one to publish in PLoS ONE. I believe the changes they have made significantly improved the quality of this paper.

Reviewer #3: I reviewed the paper. It was a good paper. The requisite modifications have been done. It can be published as it is.

7. PLOS authors have the option to publish the peer review history of their article (what does this mean?). If published, this will include your full peer review and any attached files.

Reviewer #1: No

Reviewer #3: No

---

## [Editor Report · Acceptance letter]

16 Aug 2021

PONE-D-20-40267R2 

Screen time and early adolescent mental health, academic, and social outcomes in 9- and 10- year old children: Utilizing the Adolescent Brain Cognitive Development ℠ (ABCD) Study 

Dear Dr. Paulich:

I'm pleased to inform you that your manuscript has been deemed suitable for publication in PLOS ONE. Congratulations! Your manuscript is now with our production department. 

Kind regards, 

on behalf of

Dr. Enamul Kabir 

Academic Editor

PLOS ONE